# Developmental and aging changes in brain network switching dynamics revealed by EEG phase synchronization

Dionysios Perdikis[1,2]☯*, Rita Sleimen-Malkoun[3]☯*, Viktor Müller[4], Viktor Jirsa[1]

**1** Aix-Marseille Univ, Inserm, INS, Institut de Neurosciences des Systèmes, Marseille, France, **2** Berlin Institute of Health at Charité, Universitätsmedizin Berlin, Berlin, Germany, Brain Simulation Section, Department of Neurology with Experimental Neurology, Charité, Universitätsmedizin Berlin, Corporate member of Freie Universität Berlin and Humboldt Universität zu Berlin, Berlin, Germany, **3** Aix-Marseille Univ, CNRS, ISM, Institut des Sciences du Mouvement, Marseille, France, **4** Max Planck Institute for Human Development, Center for Lifespan Psychology, Berlin, Germany

☯ These authors contributed equally to this work.
* dionysios.perdikis@bih-charite.de (DP); rita.sleimen-malkoun@univ-amu.fr (RS-M)

## Abstract

Adaptive behavior depends on the brain's capacity to vary its activity across multiple spatial and temporal scales. Yet, how distinct facets of this variability evolve from childhood to older adulthood remains poorly understood, limiting mechanistic models of neurocognitive aging. Here, we characterize lifespan neural variability using an integrated empirical-computational approach. We analyzed high-density EEG cohort data spanning 111 healthy individuals aged 9–75 years, recorded at rest and during a passive and an attended auditory oddball stimulation task. We extracted scale-dependent measures of EEG fluctuation amplitude and entropy, together with millisecond-resolved phase-synchrony networks in the 2–20 Hz range. Multi-condition partial least squares decomposition analysis revealed two independent lifespan trajectories. First, slow-frequency power, variance, and complexity at longer timescales declined monotonically with age, indicating a progressive dampening of low-frequency fluctuations and large-scale coherence. Second, the temporal organization of phase-synchrony reconfigurations followed an inverted U-shaped trend: young adults exhibited the slowest yet most diverse switching—characterized by low mean but high variance and low kurtosis of jump lengths at 2–6 Hz, and the opposite pattern at 8–20 Hz—whereas children and older adults showed faster, more stereotyped dynamics. To mechanistically account for these patterns, we fitted a ten-node phase-oscillator model constrained by the human structural connectome. Only an intermediate, metastable coupling regime qualitatively reproduced the empirical finding of maximally heterogeneous synchrony dynamics observed in young adults, whereas deviations toward weaker or stronger coupling mimicked the children's and older adults' profiles. Our results demonstrate that development and aging entail changes in the switching dynamics of EEG phase synchronization by differentially

**Data availability statement:** All measures' and simulations' data and code used to produce the results and figures are available on Zenodo at DOI: 10.5281/zenodo.15776182.

**Funding:** RSM and DP were supported by A*Midex (Aix-Marseille Initiative of Excellence) through project 'Coord-Age' (#ANR-11-IDEX-0001-02). VJ acknowledges the support of EU's Horizon Europe Programme under the Specific Grant Agreement No. 101147319 (EBRAINS 2.0 Project), Specific Grant Agreement No. 101137289 (Virtual Brain Twin Project), the support of Agence Nationale de la Recherche under France 2030, bearing the reference ANR-24-RRII-0005, on funds administered by Inserm. The funders had no role in study design, data collection and analysis, decision to publish, or preparation of the manuscript.

**Competing interests:** The authors have declared that no competing interests exist.

sculpting stationary and transient aspects of neural variability. This establishes time-resolved phase-synchrony metrics as sensitive, mechanistically grounded markers of neurocognitive status across the lifespan.

---

## Author summary

Brain activations fluctuate and synchronize according to functionally relevant patterns, both at rest and during task performance. Here, we investigated how these fluctuations change from childhood to older adulthood and what these changes reveal about healthy development and aging. Using electrophysiological brain activity recordings from 111 participants aged 9–75 years during rest and a cognitive task, we discovered two key developmental trajectories. First, a linear trend reflecting mainly a gradual weakening of slower brain rhythms with age. Second, an inverted U-shaped trend suggesting that neural flexibility peaks in young adults, who switch between neural synchronization states in uniquely diverse yet controlled ways, whereas children and older adults showed more rigid and predictable patterns. To explain this, we developed a computational model. The simulated data confirmed that young adults operate in a "sweet spot" of connectivity—balanced enough to support dynamic coordination without becoming unstable or overly rigid—unlike in childhood and older age, when connectivity becomes either too weak or too strong. Our work shows that age fundamentally reshapes how flexibly brain regions communicate. This provides sensitive markers for tracking brain health across the lifespan.

## Introduction

Neurodevelopment and aging form a continuum of structural rewiring and functional reconfiguration that reshapes performance and adaptability from childhood to late adulthood. Yet because childhood, adulthood, and older adulthood are often examined in isolation, an integrated lifespan perspective remains elusive. Rapid advances in structural and functional imaging, high-density electrophysiology and computational modeling now let us probe neural networks with unprecedented spatiotemporal resolution and link their dynamics to behavior, age, and functional status [1–3]. Leveraging these tools, the present study combines resting-state and task-based electroencephalography with generative modeling to trace how core features of brain dynamics evolve across the full spectrum of maturation, adulthood, and aging.

We focus on moment-to-moment variability and synchrony in the EEG because the brain, like any complex multiscale system, exhibits "one-to-many" and "many-to-one" mappings between activity patterns and behavior [4–6]. The resulting fluctuations are not mere noise; they are stochastic excursions around an underlying deterministic manifold that encode biologically meaningful information [7–9]. As activity unfolds, these excursions trace low-dimensional trajectories—Structured Flows on Manifolds

(SFM) [10–12]—whose geometry operates across multiple timescales and underpins distinct cognitive and sensorimotor functions [13]. Development and aging should therefore, remodel the shape of these flows in coordination with behavioral change [14]. Independent of age, each individual possesses a finite yet plastic repertoire of neuro-behavioral states, constrained by network architecture and sculpted by experience, that can be flexibly reconfigured to meet environmental demands. Treating variability as a window onto these structured flows provides a unified framework that links bottom-up structural constraints with top-down functional adaptation across the lifespan.

At the structural level, we now understand far more about how the brain wires and rewires itself to acquire new skills, maintain function, and remain adaptable across the lifespan [15,16]. Parallel advances in experimental design, quantitative analysis, and large-scale modeling have turned moment-to-moment neural dynamics into a powerful lens on brain and behavior. Function can be captured in the patterns of activity expressed within and between networks—and across states from rest to task—revealing how circuits integrate, segregate, and reconfigure [17–24]. Whether evoked by a task or emerging spontaneously, these dynamics carry the imprint of each person's neurobehavioral repertoire and their capacity to switch efficiently between functional modes.

Most research tracks development and aging in separate cohorts [25–30], with only a handful covering the full lifespan [31–35]. Theoretical frameworks converge on a noise-driven view in which stochastic perturbations expand the brain's repertoire and push healthy systems toward a metastable "sweet spot" that maximizes integration–segregation balance [36–38]. Empirically, this regime appears as variability that is midway between randomness and rigidity and often predicts higher performance and more adaptable behavior [9,39–42]. EEG studies exploit its millisecond resolution to capture such dynamics, but scalp signals blur spatial detail; multivariate and multiscale analyses therefore play a key role in disentangling local versus global processing [43–45]. Ultimately, understanding lifespan change requires linking these functional signatures to their structural underpinnings—a relationship that is far from linear given the complex, bidirectional interactions between anatomy and dynamics [46–48].

Although late adulthood is marked by a well-documented decline in white-matter integrity and inter-regional connectivity, lifespan change is not captured by a single linear descent. Instead, structural maturation and senescence follow two archetypal trajectories—one monotonic, the other non-monotonic—whose expression depends on the region or network examined [15,16,49]. Normalized gray-matter volume, along with most subcortical nuclei, decreases steadily from childhood onward, whereas the amygdala remains relatively stable, and the hippocampus traces a pronounced inverted U trajectory—maturing later and declining earlier. Douaud and colleagues [16] further identified an inverted-U component that peaks near 40 years and implicates transmodal cortical regions that both develop and degenerate ahead of the rest of the brain, rendering them especially vulnerable to age-related pathology. Similar midlife apices have been observed in graph-theoretic metrics of structural networks [50] and in white-matter tractography [15,49].

Functional connectivity exhibits the same duality. Linear declines coexist with nonlinear, inverted-U profiles whose timing and magnitude are system-specific [49,51]. A recent large-scale analysis [52] located inflection points around mid-life: global mean connectivity peaks near 40 years, while its variance tops out a decade earlier. Primary sensorimotor circuits reach maturity sooner and begin to wane earlier than association networks that support complex cognition, which in turn show greater susceptibility to later-life decline.

Since monotonic and non-monotonic trends of change are found in different structures and in interregional functional connectivity of the brain, they may be associated with different control mechanisms and neural dynamics with implications in different functional domains. Indeed, similar trends were also found in behavioral studies, although evidence is sparser and more scattered regarding lifespan changes [53–57]. This is also the case regarding changes in brain activations that are more commonly studied separately during early development (infancy-childhood, e.g., [45]), maturation (infancy/childhood-adulthood, e.g., [58]) and aging (adulthood-old age, e.g., [29]).

Because development and aging are seldom tracked within a single experiment, lifespan inferences have largely been drawn post hoc by stitching together separate studies that presume a common theoretical footing [59]. Only a handful

cover childhood through old age in one design [34,60,61], and even these are hard to compare because they deploy heterogeneous metrics—linear or nonlinear, univariate or multivariate—while variously treating EEG as stationary or explicitly time-varying. Traditional work emphasizes static, linear indices such as power spectra or peak-alpha frequency [62,63], whereas more recent studies exploit nonlinear and time-resolved approaches—fractal dimension, multiscale entropy, or sliding-window connectivity—to capture the evolving complexity of brain activity across age [41,64–66]. However, these two different approaches have typically been pursued in isolation, such that static and dynamic measures, and linear and nonlinear metrics, have not been systematically integrated, especially not within the same lifespan study. As a result, we still lack a unified empirical account of how monotonic and non-monotonic changes in neural variability relate to each other and to underlying mechanisms across development and aging. This methodological diversity underscores the need for integrative, cross-sectional analyses that span the full developmental arc and are anchored in mechanistic models.

Here, we analyze EEG data in four age groups: younger children (*YC*, mean age = 9.9 years), older children (*OC*, mean age = 12 years), younger adults (*YA*, mean age = 22.7 years), and older adults (*OA*, mean age = 67.8 years). We capitalize on both resting-state and task-evoked EEG activations to show that neural variability follows both monotonic decline and inverted U-shaped trajectories, echoing known structural and connectivity changes. Stationary features—those assumed to be stable across the time of the EEG recording—are quantified with linear (spectral power and standard deviation) and nonlinear (multiscale entropy) metrics, whereas non-stationary features are captured through time-resolved phase-synchronization dynamics, using multivariate statistics in all cases. The use of such a battery of measures is motivated by a hypothesis, associated with an inverted U-shape trajectory across age groups, that the healthy, mature, adult brain balances optimally the stability of a variety of dynamical modes and the flexibility to switch among them. Such dynamics is more likely to be evaluated by dynamic metrics, which focus on the transitions among distinct modes, than by static ones that rely on averaging across time, since no brain measurement is long enough to allow for any time-averaged statistic to capture the structure of the total state space. Accordingly, nonlinear metrics are expected to be more sensitive to such non-stationary dynamics, since transitions between functional modes are a definite signature of nonlinear dynamics. Finally, we demonstrate that a network of coupled phase oscillators reproduces these non-stationary and nonlinear dynamical signatures, providing a mechanistic framework for understanding how brain variability is modulated by age—and, by extension, by disease. The adopted integrative data analysis approach, together with the modeling strategy, extends prior work by linking static and dynamic EEG markers, linear and nonlinear metrics, as well as empirical and generative levels of description in a unified account of development and aging effects on brain signal variability and dynamics.

## Methods

### Ethics statement

The study has been approved by the ethics committee of Saarland University and has therefore been performed in accordance with the ethical standards laid down in the 1964 Declaration of Helsinki. All participants volunteered for this experiment and gave their written informed consent prior to their inclusion in the study. In the case of children, the parents did give consent for their children to participate in the study. Details that might disclose the identity of the subjects under study were omitted.

### Participants

The studied sample consisted of 24 younger children (*YC*, mean age = 9.9, SD = 0.6, age range = 9.0–10.8 years, 13 females), 28 older children (*OC*, mean age = 12.0, SD = 0.6, age range = 11.0–12.8 years, 14 females), 31 young adults (*YA*, mean age = 22.7, SD = 1.6, age range = 18.8–25.1 years, 14 females), and 28 old adults (*OA*, mean age = 67.8, SD = 3.0, age range = 63.9–74.5 years, 14 females).

## Procedure

The EEG measurement began with a 3-minute resting state recording (1.5 minutes with eyes closed, and 1.5 minutes with eyes open) and was followed by the auditory oddball task performed with eyes closed. During the task, participants heard two different tone beeps: a frequent (probability of 0.8) 1000 Hz tone as a standard stimulus and a rare (probability of 0.2) 800 Hz tone as a deviant stimulus. Stimuli were presented binaurally at 70 dB SPL with duration of 70 ms (including 10 ms rise and fall time) and inter-stimulus intervals ranging from 1200 to 1500 ms. There were two different experimental conditions: passive listening (unattended) and active counting (attended). In the first condition, participants merely listened to the tone beeps without any response, whereas in the second condition, they had to attend to stimuli, count the deviant tones and report their number after the end of the session. For this study we considered three conditions, all with eyes closed: resting state (*REC*, i.e., *Rest with Eyes Closed*), auditory oddball task without counting (*UOT*, i.e., *Unattended Oddball Task*) and auditory oddball with counting (*AOT*, i.e., *Attended Oddball Task*). The condition of resting state with eyes open was not included since it differed largely in its frequency content compared to all other conditions, which interfered with tasks contrasts. Instead, we focused on studying task differences under comparable conditions along the axis of increasing cognitive and task demands.

## EEG recordings and preprocessing

The electroencephalogram (EEG) was recorded with a sampling rate of 500 *Hz* in a frequency band ranging from 0.5 to 100 *Hz*. The electrodes were placed according to the international 10–10 system. The left mastoid was used as a reference, and the right mastoid was recorded as an active channel. The data were re-referenced off-line to an average of the left and right mastoids for further analysis. Vertical and horizontal electrooculograms were recorded for control of eye blinks and eye movements. Eye movement correction was accomplished by independent component analysis [67]. Thereafter, artifacts from head and body movements were rejected by visual inspection. Data were downsampled to a sampling rate of 250 *Hz*, segmented in artifact free 10 s segments (i.e., comprising $N_t = 2500$ data points each), and mean centered within segments before further analysis. Table 1 shows the statistics of the resulting number of segments included in the analysis for each condition and group.

A more detailed description of the participants' cohort, the experimental procedure and the EEG recording and preprocessing can be found in [33,34,41].

We proceed with describing the analysis pipeline leading to the various metrics used in this study. The pipeline is sketched out in Fig 1 for the characteristic example of a data segment of a young adult at rest with eyes closed.

**Table 1. Mean, standard deviation, minimum and maximum numbers of EEG segments per group and condition included in the analysis.**

| Groups | Conditions | Mean | Standard Deviation | Minimum | Maximum |
|---|---|---|---|---|---|
| YC | REC | 7.0 | 1.3 | 4 | 8 |
| | UOT | 21.2 | 3.4 | 14 | 25 |
| | AOT | 22.6 | 1.9 | 19 | 25 |
| OC | REC | 7.6 | 0.8 | 5 | 8 |
| | UOT | 22.8 | 2.0 | 18 | 25 |
| | AOT | 22.9 | 2.0 | 18 | 25 |
| YA | REC | 7.8 | 0.6 | 5 | 8 |
| | UOT | 24.0 | 2.0 | 15 | 25 |
| | AOT | 23.0 | 3.1 | 11 | 25 |
| OA | REC | 7.4 | 1.1 | 12 | 25 |
| | UOT | 22.3 | 3.4 | 12 | 25 |
| | AOT | 21.6 | 3.1 | 15 | 25 |

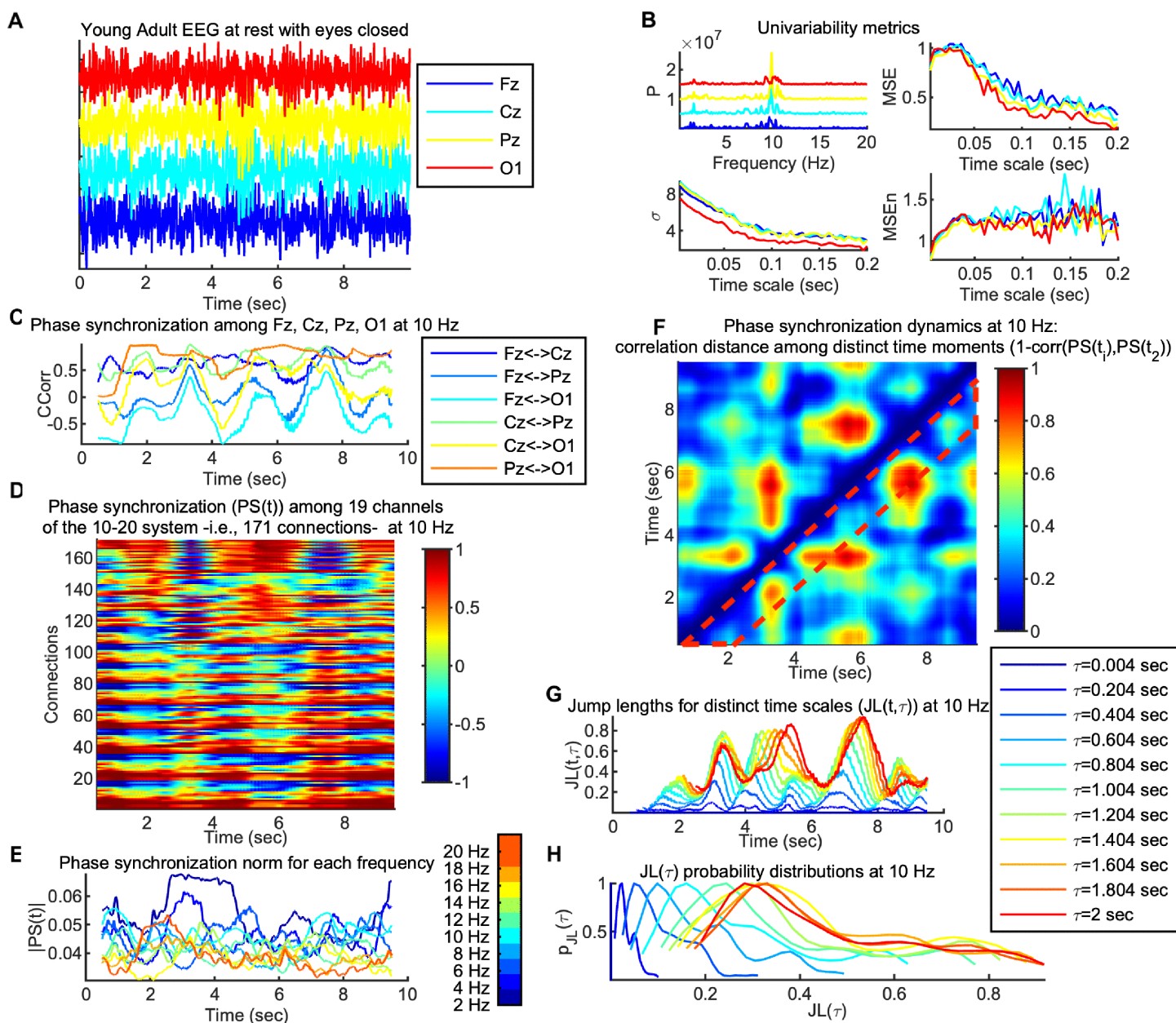

**Fig 1. Data analysis pipeline exemplified.** *Panel A*: Time series of four channels (*Fz, Cz, Pz, O1*, out of 58 in total of the international 10–10 system) of a characteristic 10 sec segment of a young adult (*OA*) participant from resting state with eyes closed (*REC*). *Panel B*: Univariate variability metrics for the data of *Panel A* (same channels and colors); *power spectrum* (*P*; top left, x-axis is frequency in Hz), *standard deviation* (σ; bottom left, x-axis is time scale in sec), *normalized multiscale entropy* (*MSEn*; bottom left, x-axis is time scale in sec) and *multiscale entropy* (*MSE*; top right, x-axis similar to *MSEn*). *Panel C: Phase synchronization* (*Circular Correlation Coefficient*; *CCorr*) of the data of *Panel A* at 10 Hz, with a sliding, fully overlapping, window of 1.004 sec. Note that there is a rich dynamic leading to a non-stationary (in the wide sense) signal. *Panel D*: Similar to *Panel C* but for 19 channels, which leads to 171 connections (y axis). *Panel E*: Normalized Euclidean norm of the *phase synchronization* networks (|*PS(t,f)*|) at each time point and frequency. Colors correspond to the different frequency bins as explained by the colorbar. The mean ($\mu_{|PS(f)|}$) and standard deviation ($\sigma_{|PS(f)|}$) for each frequency across time were computed for statistical comparisons. *Panel F: Phase synchronization dynamics* (*PSD*) at 10 Hz computed as the correlation distance among the phase synchronization (*PS*) networks of *Panel D*, at every two distinct time points, leading to a time versus time plot. Note the rich dynamic, according to which there is an alternation of time intervals with small distance (in bluish colors) versus large distance (in reddish colors) changes in the topology of the *PS* networks. The area surrounded by the dotted red line indicates approximately the timescales (diagonals of the *PSD* matrix) that were chosen for the statistical analysis, in the range 1.004-2 sec. *Panel G*: Considering *PSD* as a random walk in the 171-dimensional space of possible *PS* networks, time sequences of *jump lengths* (*JL(t,τ)*) for selected timescales τ are plotted, for the indicated time scale range, i.e., the

corresponding diagonals of the *PSD* matrix (see the legend for the exact timescales shown in different colors). *Panel H:* Probability distributions of the *JL* sequences ($p_{JL(\tau, 10\,Hz)}$) of *Panel G*, computed at 25 equidistant points, and normalized with the maximum value, only for demonstration. The sample mean ($\mu_{JL(\tau,f)}$), standard deviation ($\sigma_{JL(\tau,f)}$) and kurtosis ($k_{JL(\tau,f)}$) of such *JL* sequences are computed for further statistical analysis. The x-axis in *Panels A*, *C-G* is the same and corresponds to time. For all phase synchronization graphs (*Panels C-G*), time points correspond to sliding window centers.

## Univariate variability metrics

The following variability metrics have been applied to all data segments and MATLAB (The Mathworks Inc.) was used for all computations. We computed Power spectrum (*P*), standard deviation ($\sigma$), Multiscale Entropy (*MSE*), and an extended version of the latter (*MSEn*), in an identical way to [41]. *MSE* is a representative metric for the complexity of a signal across a large range of temporal scales [68,69]. The *MSE* algorithm combines the computation of *sample entropy* (*SampEn*) as introduced in [70] with a coarse graining procedure that acts like downsampling with a Finite Impulse Response (FIR) low-pass filter, in order to compute this entropy for different timescales. By normalizing with the standard deviation at each time scale in the case of *MSEn*, we aim at controlling for the confounding effect of decreasing standard deviation with increasing coarse-graining and, thus, improving the interpretability of our results. A previous study [41] has shown that this modified metric provides additional information on EEG variability (see also [71]) for a similar approach), focusing more on nonlinear properties of the underlying dynamics than *MSE*. Characteristic examples of these four metrics for channels Fz, Cz, Pz, and O1 are depicted in Fig 1, *Panel B*.

## Phase synchronization metrics

For the phase synchronization analyses, we selected $N_{ch} = 19$ channels out of the available 58, based on the 10–20 system (see S1 Table in Supplementary Information), to avoid volume conduction effects between electrode sites located close together. We then applied to all data segments $x(t)$ a continuous Morlet wavelet transform with cycle number $c = 7$:

$$WT(t, f) = \frac{1}{\sqrt{f}} \int x(t) A_f e^{\frac{-t^2}{2\sigma_t^2}} e^{i2\pi ft} dt,$$

where $i = \sqrt{-1}$, $f$ is the center frequency that spanned 10 values from 2 to 20 Hz in steps of 2 Hz, $\sigma_t = \frac{c}{2\pi f}$, and $A_f$ is a normalization constant, chosen such as to retain the signal's amplitude values for a given time-frequency point. Thus, the wavelet duration was given by $2\sigma_t = \frac{c}{\pi f}$, its wavelength by $WL = 2\pi\sigma_t$ and its spectral bandwidth by $2\sigma_f$, where $\sigma_f = \frac{1}{WL} = \frac{1}{2\pi\sigma_t} = \frac{f}{c}$. Then, we extracted the phase $\varphi(t, f)$ from the complex $WT(t, f)$ and computed *Circular Correlation Coefficient* (*CCorr*; see [72]), as a measure of *phase synchronization* (*PS*), among all $N_{con} = N_{ch}*(N_{ch}-1)/2 = 171$ pairs of channels for each data segment, within a sliding, full-overlapping, window of length 1.004 seconds or 251 data points. CCR was computed as:

$$CCorr_{i,j}(t, f) = \frac{\sum_{t-125}^{t+125} \sin\left(\varphi_i(t, f) - \overline{\varphi_i}\right) \sin\left(\varphi_j(t, f) - \overline{\varphi_j}\right)}{\sqrt{\sum_{t-125}^{t+125} \sin^2\left(\varphi_i(t, f) - \overline{\varphi_i}\right) \sin^2\left(\varphi_j(t, f) - \overline{\varphi_j}\right)}},$$

where $i$, $j$ denotes here indexes of two different channels, the time variable $t$ takes integer values and corresponds to the center of the sliding window, and $\overline{\varphi_{ij}}$ denotes the sample mean of the phase within a sliding window. *CCorr* takes values between 1 (for perfect in phase synchronization) and -1 for perfect antiphase -180°- synchronization), whether small values around 0 signify orthogonality of the two signals as a lack of synchronization. Thus, this measure evaluates phase covariance between the two signals and not phase difference constancy (as for instance phase locking statistics [73]), being a suitable measure of EEG functional connectivity. Fig 1, *Panel C* depicts characteristic examples of *CCorr*

time series for all combinations of channels Fz, Cz, Pz, and O1 at 10 Hz and *Panel D* for the whole network of $N_{con}$ connections.

**Phase synchronization norm.** Subsequently, we computed for each time point the normalized Euclidian norm of the phase synchronization across all connections among channels, as:

$$\left|PS(t,f)\right| = \frac{1}{N_{con}}\sqrt{\sum_{k}^{N_{con}} CCorr_k^2(t,f)},$$

where the index $k$ spans all $N_{con}$ connections (i.e., we treat a synchronization network $PS(t,f)$ at a particular time-frequency point as a vector of $N_{con}$ elements). Then, we computed the mean ($\mu_{|PS(f)|}$) and standard deviation ($\sigma_{|PS(f)|}$) of this quantity across time for each data segment. Thus, we obtained measures of the mean and variability of the norm (magnitude) of the phase synchronization networks across time for each frequency. Fig 1, *Panel E* depicts characteristic *PS* time series for all frequencies.

**Phase synchronization dynamics.** Next, we evaluated *phase synchronization dynamics* (*PSD*), i.e., the fluctuations of those synchronization networks across time, focusing on the changes only on their topology. For that purpose, we calculated correlation distance among every pair of synchronization networks at time points separated in time by a time scale factor $\tau$:

$$PSD(t, t+\tau, f) = 1 - corr\left(PS(t,f), PS(t+\tau, f)\right),$$

where operator *corr*(.) corresponds to the correlation coefficient, which is independent of the magnitude (norm) of $PS(t,f)$. Fig 1, *Panel F* depicts a characteristic example of a PSD matrix at 10 Hz. Treating *PSD* as a random walk in the $N_{con}$th space of possible synchronization networks (*PS*), we selected timescales in the range $\tau \in [1.004, \ 2]$ sec, i.e., spanning almost 1 sec of non-overlapping windows, using all possible steps of 0.004 ms. Thus, we defined fluctuations' *jump lengths* $JL(\tau,f)$, as the (correlation) distances *PSD* have moved in $\tau$ time for all time points $t$, following:

$$JL(\tau, f) = \left\{PSD(t, t+\tau, f)\right\}, \ t \in [0.5\tau + 0.004, \ 9.5] \ s,$$

i.e., each $JL(\tau,f)$ vector of data points corresponds to the $\tau$-th diagonal of the $PSD(t, t+\tau, f)$ matrix. Characteristic $JL(\tau,f)$ vectors for $f = 10$ Hz and a few selected timescales $\tau$ are plotted across time $t$ in Fig 1, *Panel G*, whereas *Panel H* depicts their probability distributions computed with MATLAB function *ksdensity()* at 25 equidistant points, only to demonstrate their long-tailed form. Finally, we computed the mean ($\mu_{JL(\tau,f)}$), the standard deviation ($\sigma_{JL(\tau,f)}$), and the kurtosis $k_{JL(\tau,f)}$ of $JL(\tau,f)$. $\mu_{JL(\tau,f)}$ can be interpreted as the average speed of *PSD*, $\sigma_{JL(\tau,f)}$ evaluates the *PSD* variability, whereas a leptokurtic distribution of $JL(\tau,f)$, as evidenced by a high $k_{JL(\tau,f)}$, characterizes *PSD*, as "switching dynamics", i.e., dynamics where very short or very long jumps are more likely than in a normal (Gaussian) distribution. The *PSD* metrics allowed us to statistically evaluate the dynamics of transitions in time among distinct phase synchronization networks, which roughly correspond to different modes of brain dynamics, as opposed to all other metrics used in this study that rely on averaging across time.

## Partial Least Squares (PLS) statistical analysis

Partial Least Squares [74,75] is a multivariate statistical method that aims at revealing the relationship between two blocks of datasets, i.e., on the one hand, the metrics computed in this study, and, on the other hand, some vectors coding for the experimental design. Thus, it is a method suitable for explorative investigation of spatial and/or time distributed signal changes by combining information across different signal dimensions. The method is based on a decomposition

of the covariance of the two blocks in a set of new variables that optimally -in a least square sense- relate them, and, therefore, explain as much of covariance with as few dimensions as possible. In this study, we used mean centering Task PLS ("mc-Task PLS"), which allows us to explore the contrast of the experimental design that optimally covaries with our metrics. The method starts by constructing a brain data matrix for each experimental group. Condition averages are taken across conditions, group matrices are concatenated, and the grand average is removed (all processing is performed in parallel for each data element). The resulting matrix undergoes a singular value decomposition. The output of the mc-Task PLS yields three matrices: i) the *task design latent variables*, i.e., the *saliences of the contrasts* describing the relations among the conditions and groups of our design for each contrast, ii) the *brain latent variables*, i.e., *element saliences* that are proportional to the covariance of each data element with each one of the task contrasts, and iii) *singular values* that are proportional to the covariance explained by each contrast (the ratio of each squared singular value by the sum of squares of all singular values of all latent variables (LVs) describes the percentage of covariance each LV accounts for). The number of resulting singular values, one for each contrast, depends on the degrees of freedom of the design. In our design we had four groups, namely younger children ('YC'), older children ('OC'), young adults ('YA') and old adults ('OA') participants, and three conditions, i.e., 'REC', 'UOT' and 'AOT' as explained above. Thus, mc-Task PLS returned, *number of conditions* x *number of groups* -1 = 3*4 − 1 = 11 orthonormal contrasts. Furthermore, we computed *brain scores* that indicate the strength of the task effect of each contrast per participant and condition. In other words, the brain score of a particular participant for a specific contrast and condition is the covariance of the brain data of this participant for that condition with the corresponding *brain latent variable* vector of the contrast in question. PLS addresses the problem of multiple comparisons for statistical significance via a permutation test and the problem of element-wise reliability via a bootstrap resampling test. The permutation test is performed on the singular values with resampling of the initial data matrices across conditions and groups without replacement. This permutation test yields a *p*-value for each *task latent variable*, i.e., for each contrast. We used an alpha value of significance equal to 0.05 throughout this study. For the bootstrap test, the initial data matrix is resampled with replacement within conditions and groups. For the contrasts we present here, we plotted the *task latent variables*, together with the mean brain scores and their intervals of 95% confidence that are derived from the bootstrap test, after mean-centering and normalizing with the respective singular value. Conditions or groups, the confidence intervals of which do not overlap, are reliably distinguished by a contrast. For the *brain latent variables*, we calculated bootstrap ratios by dividing each element with its standard error as computed by the corresponding bootstrap sample distribution. Bootstrap ratios greater than 2.5758 approximate the 99th two-tailed percentile for a particular element (see Z-score table).

## Simulation

We defined a network of $N = 10$ phase oscillators, coupled mutually via a coupling function derived from synaptic couplings [76]. The model system reads

$$\frac{d\theta_i}{dt} = \omega \left( 1 - a \cos \theta_i + (1 - \cos \theta_i) \left( \frac{K}{N} \sum_{\substack{j = 1, \\ j \neq i}}^{N} w_{ij} \cos \theta_j \right) + Q\eta_i(t) \right),$$

where $\theta_i$ is the phase state variable of each oscillator $i$, $\omega = 2\pi f$ with $f = 10$Hz is the central angular frequency, $a$ is a parameter controlling the stability of the system (stability decreases for increasing $a$), $K$ is the global coupling strength parameter, and $w_{ij} = w_{ji} \geq 0$ is the symmetric connectivity weight between each pair of oscillators in the network, whereas $\eta_i(t)$ is white Gaussian noise with strength $Q = 0.01$. For the connectivity weights $w_{ij}$ we used a publicly available The

Virtual Brain [77] dataset with values corresponding to numbers of white matter tracts, derived from Diffusion Tensor Imaging tractography. We selected a sub-circuit of the first $N$ = 10 nodes and normalized the connectivity matrix by dividing them with their 95$^{th}$ percentile to get the resulting weights $w_{ij}$ (see S1 Fig in the Supplementary Information). We performed simulations for different combinations of parameters' values $a \in \{0.0,\ 0.1,\ 0.25, 0.5, 0.75, 0.9, 0.95,\ 1.0\}$ and $K \in \{0.0,\ 0.1,\ 1.0, 2.0, 5.0, 10.0, 100.0\}$. For each simulation, we integrated the system with the Euler scheme and a time step of dt = 10 ms for 11 sec, removing the first 1 sec to discard transients. Then we applied to the resulting $\theta_i(t)$ time series a similar analysis to the EEG data, namely, after downsampling to 250 Hz, we computed the phase synchronization time series $CCorr_{i,j}(t, 10\ Hz)$ with a sliding, partly-overlapping, window of length 1 sec (250 data points) in steps of 100 ms, and, subsequently, the phase synchronization dynamics matrices $PSD(t, t + \tau, 10\ Hz)$ with $\tau \in [1,\ 2]$ sec. Fig 2 depicts selected time series (panel A; from top to bottom $\cos(\theta_i(t))$, $\cos(\theta_i(t) - \omega t)$ for the phase dynamics to be revealed and $CCorr_{i,j}(t, 10\ Hz)$) and $PSD$ matrices (panel B) for characteristic points of the above two dimensional parameters' space. Finally, for the intermediate pair of parameters' values $a = 0.5,\ K = 2.0$ we clustered the transient phase synchronization networks based on the $PSD$ matrix as a (correlation) distance metric among time windows. For this purpose, we used the function *cluster.kmedoids* of the Python3 package *pyclustering* [78]. The selection of the specific pair of parameters, as well as the number of clusters (three), and the initial medoids, as inputs to the clustering algorithm, were based on visual inspection of the respective $PSD$ matrix. Then, we plotted $PSD$ as a 3-dimensional trajectory (Fig 2, panel C), for which the position of each point (corresponding to the phase synchronization network at time window $t$) depicts the correlation $1 - PSD(t, m_k, 10\ Hz)$ to each one of the clusters medoids' synchronization networks, where $m_k$ is the medoid of cluster $k$ for $k = \{1, 2, 3\}$. Note that, given that the self-correlation distance $PSD(m_k, m_k, 10\ Hz) = 0$, it follows that $1 - PSD(m_k, m_k, 10\ Hz) = 1$.

## Results

### Univariate variability metrics

The mean curves with standard error intervals of Power spectrum ($P$), standard deviation ($\sigma$), Multiscale Entropy ($MSE$), and an extended version of the latter ($MSEn$) are shown in Fig 3, allowing for comparisons between age groups under the same task condition. The mc-Task PLS analyses, to which we turn next, revealed one significant latent variable (LV) for $\sigma$ and $MSE$ and two such variables for $P$ and $MSEn$, according to the permutation test. The first LV for all univariate variability metrics was significant ($p < 0.001$) and explained most of the covariance (approximately 71.8%, 97.6%, 96.4 and 83.4% for $P$, $\sigma$, $MSE$ and $MSEn$, respectively). As can be seen in Fig 4 through the task LV and normalized brain scores' graphs, this LV reliably distinguishes children from adults for all four metrics with negligible condition differences (i.e., no strong task-related effect is detected). Statistical trends, although not significant, exist between $YC$ and $OC$ for $MSE$ and $YA$ and $OA$ for the three other metrics (stronger for $MSEn$). The direction of all these effects is the same both for aging and development, sketching, in general, a monotonic change with age across lifespan. We observe that adults have generally less power than children (see bootstrap ratios of the brain LV in Fig 5), an effect that is stronger for fronto-central channels and frequencies below 10 Hz. For frequencies above 20 Hz, the effect diminishes or even reverses for some fronto-central channels (e.g., $C6$, $F8$, $F7$), but with statistically unreliable bootstrap ratios. Accordingly, adults have a lower standard deviation than children for all timescales, the effect being stronger for more anterior channels and longer timescales. As for the entropic metrics, adults exhibit higher (lower) $MSE$ for short (long) timescales than children, the crossing time scale point being around 20 ms, which relates to the notch filter around 50 Hz. This effect is also stronger for fronto-central channels. $MSEn$ follows closely the results of $MSE$, but with a few notable differences —beyond the marked aging effect between $YA$ and $OA$ described above. Specifically, the $MSEn$ effect is stronger for central channels, and the crossing point is at longer timescales (around 50 ms for anterior channels and 40 ms for posterior ones). It is also worth noting that normalizing with the magnitude of variability (i.e., $\sigma$) weakens the effect of the notch filter in comparison with $MSE$. As for the second LV of the $P$ and $MSEn$ that was significant ($p < 0.003$ and $p < 0.015$, respectively) and explained approximately 18.9% and 9.7% of the covariance, respectively, it mainly contrasts $OC$ and $YA$ to $YC$ and $OA$, i.e., it sketches an inverted

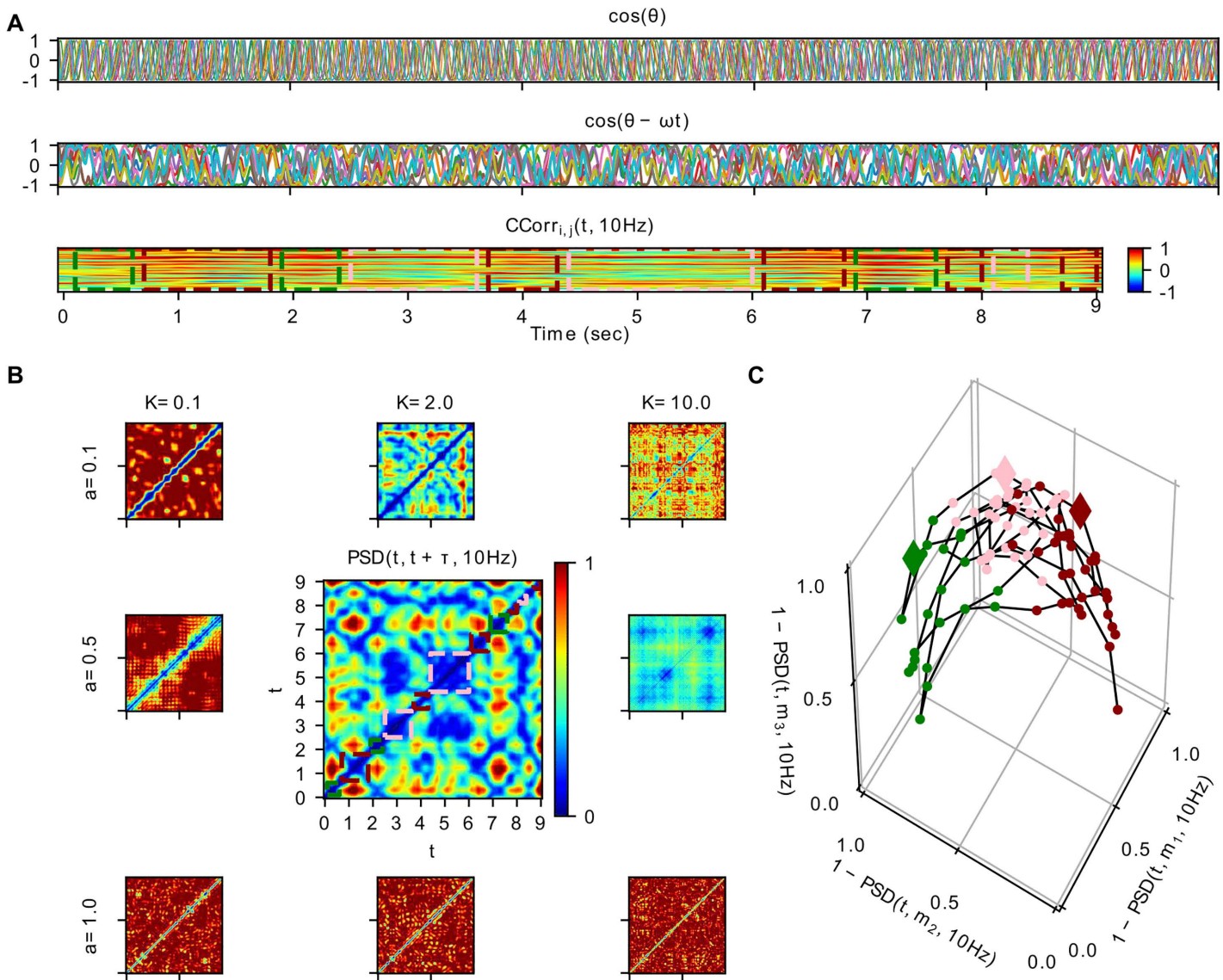

**Fig 2. Simulations for illustration of the theoretical concepts used to interpret the results.** In this figure, we show how a toy model of ten coupled phase oscillators (of central frequency $f = 10$ Hz) can exhibit non-stationary phase synchronization dynamics, which evolves along distinct network "states" on a slow time scale. We present simulations for different values of the parameters of stability $a$ (for values {0.1, 0.5, 1.0}; stability decreases for increasing $a$), and global coupling scaling $K$ (for values {0.1, 2.0, 10.0}). *Panel A:* From top to bottom, we plot for $a = 0.5$, $K = 2.0$ the simulation time series of (a) the cosine of phase $\theta_i(t)$ and (b) the cosine of $\theta_i(t) - 2\pi ft$, as well as (c) the phase synchronization ($CCorr_{i,j}(t, 10Hz)$), with a sliding window (of 1 sec, in steps of $\tau = 100$ ms). In plots (a-c) $t$ stands for time (points for (a,b), and windows for (c)), and $i, j \in [1,10]$ are the indices of the phase oscillators for each synchronization index (plotted in distinct colored lines). For the bottom plot (c), warm (cold) colors stand for stronger positive (negative) synchronization (respectively; see also the color bar on the right), whereas rectangular dotted lines depict time intervals, the synchronization networks of which, have been classified to three distinct clusters (see below), each one of a different color (green, dark red, pink; no rectangles are depicted for windows smaller than 200 ms, i.e., two subsequent sliding steps). The middle plot (b), for which the linear central frequency phase increment has been subtracted from the phase time series, serves to reveal the slow modulation of the phases $\theta_i(t)$. *Panel B:* We show Phase Synchronization Dynamics (*PSD*) of $CCorr_{i,j}(t, 10Hz)$, for the different values of parameter $a$ ($K$), changing in the vertical (horizontal) direction (respectively). The color of each pixel of the $PSD(t, t+\tau, 10Hz)$ images depict the correlation distance between the synchronization networks of two distinct time windows. Therefore, both axes depict time (windows), and each diagonal pixel line away from the central one corresponds to a distinct time scale $\tau$. At the central panel with $a = 0.5$, $K = 2.0$ note the qualitative similarity with the EEG data PSD matrix of Fig 1. The rectangular, colored dotted lines along the central diagonal enclose time windows that correspond to the distinct clusters of synchronization networks. *Panel C:* We demonstrate the synchronization dynamics for $a = 0.5$, $K = 2.0$ as a stochastic exploration of a 3-dimensional space spanned by characteristic networks. We clustered the transient phase synchronization networks with a *k*-medoids algorithm using the *PSD* matrix as a (correlation) distance metric (see *Methods*). Thus, the positions of the plotted points depict the

correlation 1 - *PSD(t, $m_k$, 10Hz)* of each network at time window *t* to each one of the three cluster medoids $m_k$ of the synchronization networks. Note that, given that the self-correlation distance *PSD($m_k$, $m_k$, 10Hz)* = 0, we have 1 - *PSD($m_k$, $m_k$, 10Hz)* = 1 for the three cluster medoids.

U-shape component with reverse effects for development and aging, with no clearly distinguishable pattern related to task conditions (see Fig 6). Turning to the corresponding brain LV (Fig 7), *OC* and *YA* exhibit more power and higher *MSEn* for a short range of frequencies (approximately 6–24 Hz) and timescales (20–50 ms), respectively, than *YC* and *OA*. The effect is stronger and wider at more posterior channels for *P*, and at more anterior ones for *MSEn*.

## Phase synchronization metrics

**Phase synchronization norm.** Fig 8 shows the mean values with standard error intervals of the time-averaged sample *mean* ($\mu_{|PS(f)|}$) and *standard deviation* ($\sigma_{|PS(f)|}$) of the *phase synchronization norm* for each frequency bin (|*PS(f)*|), plotted across age groups under the same task condition. The respective mc-Task PLS analyses revealed only one significant LV for each metric ($p < 0.001$) that explained approximately 95.6% of the covariance for $\mu_{|PS(f)|}$ and 86% for $\sigma_{|PS(f)|}$. The LV for $\mu_{|PS(f)|}$ reliably contrasts adults to children, with an additional—but statistically unreliable—difference between *YC* and *OC*. In contrast, the LV for $\sigma_{|PS(f)|}$ contrasts reliably *OA* to *YC*, while OC and YA fall in between, without

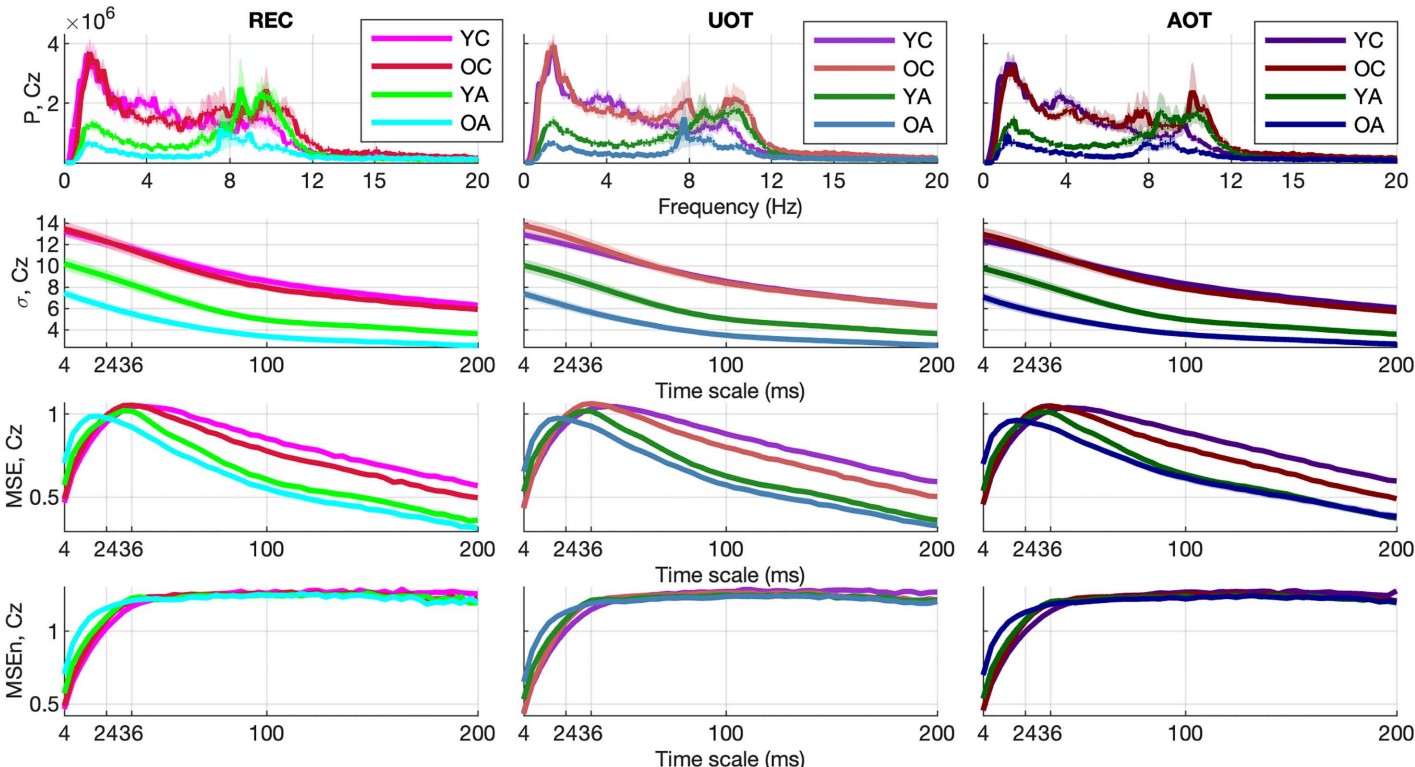

**Fig 3. Group and condition's means and standard error intervals of univariate variability metrics.** From top to bottom: *power spectra (P)*, *standard deviation (σ)*, *multiscale entropy (MSE)*, and *normalized multiscale entropy (MSEn)*, for channel *Cz*, are shown for all conditions from left to right columns ('YC' –magentish colors, 'OC' –reddish colors, 'YA' –greenish colors, and 'OA' –blueish colors) with darker colors for increasing attentional demands ('REC', 'UOT', 'AOT'). Thick lines and areas of faded colors represent the means and the standard error intervals, respectively. Horizontal axes depict frequency for *P*, and timescale for *σ*, *MSE* and *MSEn*.

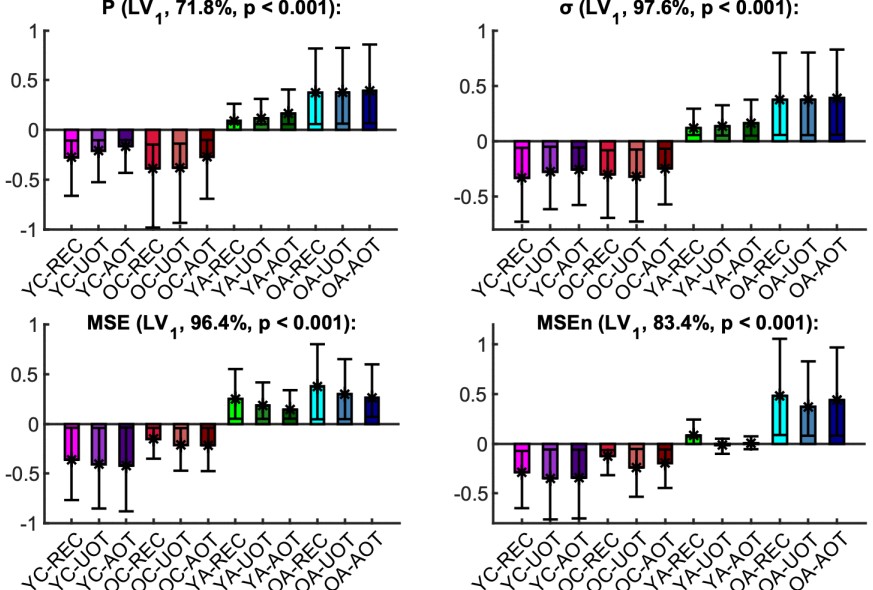

**Fig 4. First task latent variable and normalized brain scores of univariate variability metrics: monotonic component.** Each panel shows the weights of the first *task latent variable* (asterisks), and the normalized brain scores with 95% confidence intervals (bars and error bars) of the metrics (*P*, top left, *σ* top right, *MSE* bottom left, *MSEn* bottom right). Each bar corresponds to a group and condition combination, with groups being arranged in increasing age from left to right, and conditions arranged in an order of increasing attention demands (i.e., from '*REC*' to '*AOT*'), also from left to right. Color conventions are identical to Fig 3. The name of each metric, together with the corresponding *p*-value (as derived from the permutation test for significance) and the percentage of covariance explained by this latent variable are shown at the title of the respective panel. Non-overlapping confidence intervals signify that conditions and/or groups are separated reliably by the contrast. Thus, we note that for all metrics except for *MSEn*, this latent variable separates mainly adults with children, whereas for *MSEn* young adults are also distinguished being in between children and old adults. In general, these latent variables sketch a monotonic change across lifespan.

being reliably differentiated from either extreme age groups (see Fig 9, top row with task LV and normalized brain scores). Both contrasts correspond to generally increasing effects across conditions for their respective metrics, that are statistically reliable across all frequencies for $\mu_{|PS(f)|}$ —particularly for frequencies up to 12 Hz —and for $\sigma_{|PS(f)|}$ within the 2–12 Hz range, excluding 10 Hz. This frequency-specific reliability is illustrated by the brain LV bootstrap ratios shown in the bottom row of Fig 9.

**Phase synchronization dynamics.** The situation is quite different, though, for the statistics of *phase synchronization dynamics* (*PSD*), i.e., of the fluctuations in the topology of the *phase synchronization* networks. Figures of means and *t*-scores of the sample *mean* ($\mu_{JL(\tau,f)}$), the *standard deviation* ($\sigma_{JL(\tau,f)}$), and the *kurtosis* ($k_{JL(\tau,f)}$) of the *jump lengths* $JL(\tau,f)$ of these fluctuations are shown in S2-S7 Figs of the Supporting Information. The first LV of the mc-Task PLS analyses was significant ($p < 0.001$) and explained approximately 96.7% of the covariance for $\mu_{JL(\tau,f)}$, 55.7% for $\sigma_{JL(\tau,f)}$, and 83.3% for $k_{JL(\tau,f)}$. This LV reliably contrasts *YA* with children across all three metrics, with *OA'* being somewhere in between; their separation from either *YA* or children is marginal and not statistically reliable (Fig 10, top row). Thus, this component exhibits an inverted U-shaped trajectory across the lifespan—reflecting opposing effects during development and aging. Turning to the brain LV (Fig 10, bottom row), we observe the following age-related patterns: *YA* exhibit lower $\mu_{JL(\tau,f)}$ values across all frequencies (the effect being more pronounced at higher frequencies); *YA* exhibit higher $\sigma_{JL(\tau,f)}$ values across frequencies in the range 2–6 Hz and lower values in 8–20 Hz (reliable only in 14–16 Hz); *YA* exhibit higher $k_{JL(\tau,f)}$ values for frequencies in the range 8–20 Hz and lower values in 2–6 Hz (reliable only in 2–4 Hz). Overall, the *PSD* of *YA* is generally slower, more

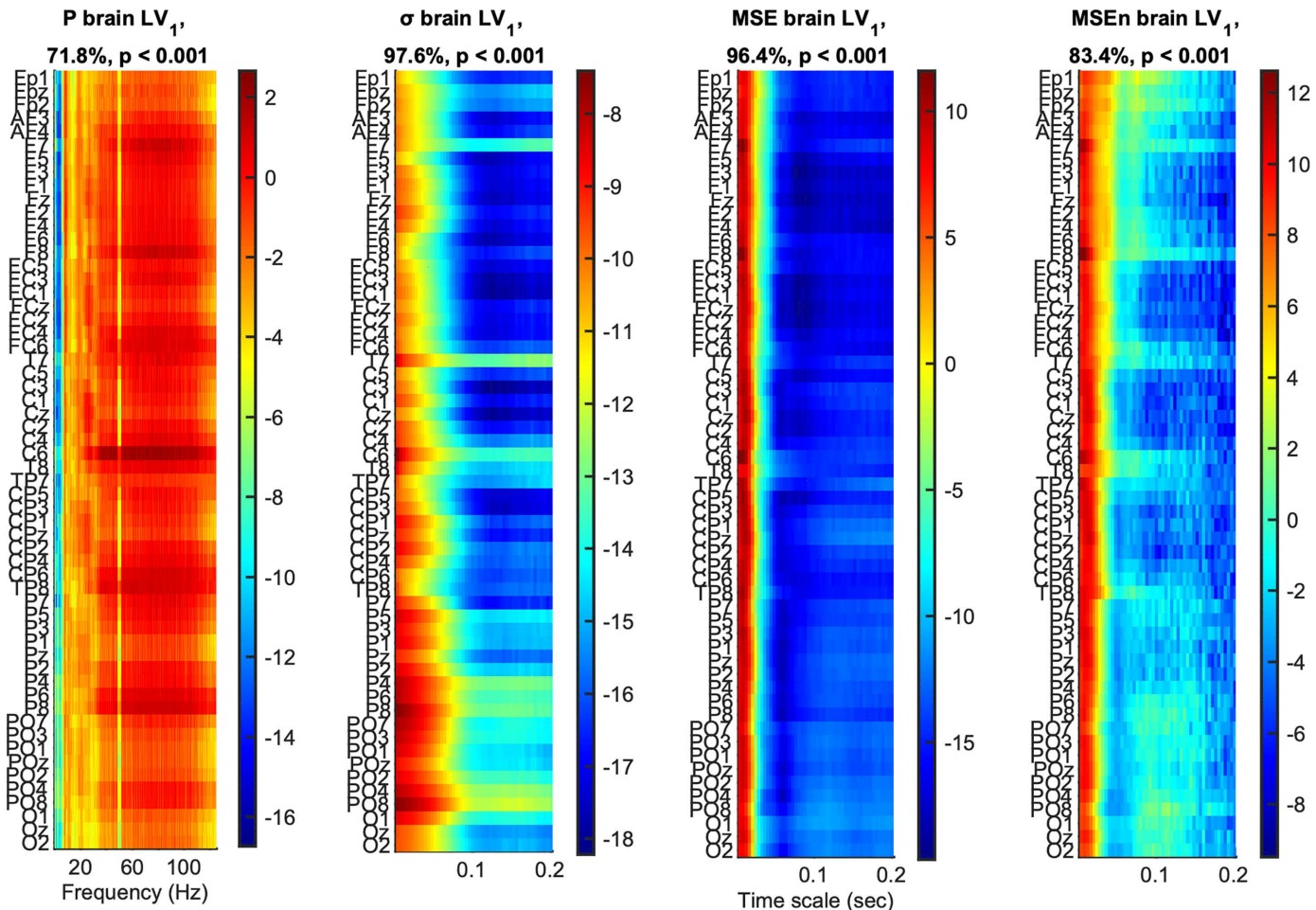

**Fig 5. First brain latent variable of univariate variability metrics: monotonic component.** The panels show how much each data *element*, i.e., a metric's data point, covaries with the task contrasts of Fig 4 for *P*, *σ*, *MSE* and *MSEn*, from left to right, in terms of bootstrap ratios. Absolute values larger than 2.5758 approximate the 99th two-tailed percentile. The vertical axis for all panels depicts channels arranged from top to bottom, starting from frontal and left hemisphere channels to occipital and right hemisphere ones. The horizontal axes depict frequency for *P* and timescale for *σ*, *MSE* and *MSEn*, whereas the titles present information on the statistical significance and the percentage of covariance explained, similarly to Fig 4. Positive (negative) bootstrap ratios signify points that exhibit a positive (negative) correlation with the task latent variables of Fig 4, i.e., they correspond to elements that were higher for adults (children, respectively).

variable at frequencies below 6 Hz, and more leptokurtic at frequencies above 6 Hz, compared to all other groups—most notably in contrast to children. All these effects appear to hold across task conditions with no distinguishable task-related differences. Finally, we identified that $\sigma_{JL(\tau,f)}$ and $k_{JL(\tau,f)}$ exhibit a second significant LV ($p < 0.003$ for $\sigma_{JL(\tau,f)}$ and $p < 0.006$ for $k_{JL(\tau,f)}$) that explained approximately 33.3% and 6.8% of the covariance, respectively (see S8 Fig in the Supporting Information).

## Simulation: Non-stationary synchronization dynamics in a network of coupled phase oscillators

Computational simulations of a network of $N = 10$ coupled phase oscillators, each with a center frequency of 10 Hz, exhibit slow (approximately on a time scale of 1 sec) intermittent fluctuations in synchronization (see Fig 1). The synchronization dynamics depends crucially on the degrees of multistability of the phasing in the network and the stability of the individual

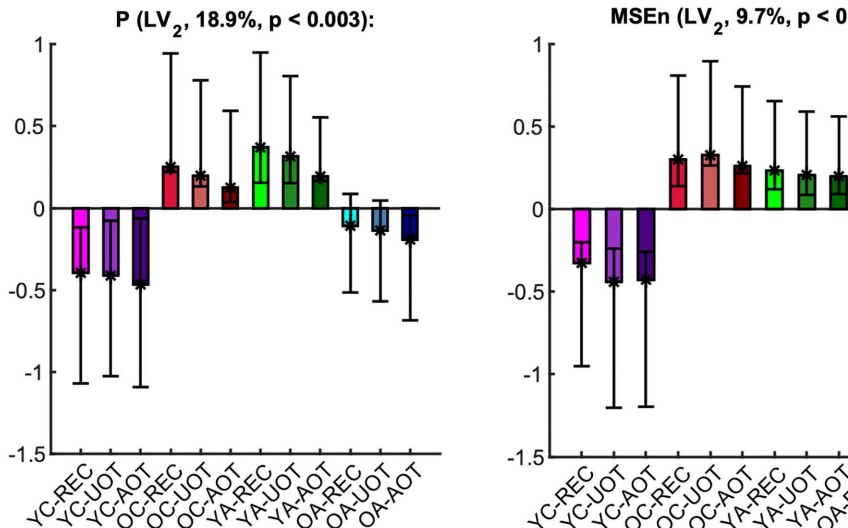

**Fig 6. Second task latent variable and normalized brain scores of univariate variability metrics.** Each panel shows the weights of the second *task latent variable* (asterisks), and the normalized brain scores with 95% confidence intervals (bars and error bars) of *P* (left) and *MSEn* (right). Color and other conventions are identical to Fig 4. An inverted U-shape change of reverse effects for development and aging, i.e., contrasting 'YC' and 'OA' to 'OC' and 'YA', is significant for *P* and *MSEn*.

attracting states. As the parameters for stability $a$ and global coupling strength $K$ alter the landscape, the degree of variability of the emergent synchronization changes systematically. For low $a$ and/or low $K$, synchronization dynamics (*PSD*) is highly suppressed as it stays close to the same attractor. For intermediate values, rapidly changing and fluid, but also persistent *PSD* appears. For high $a$ and/or $K$, the *PSD* saturates, exhibiting too fast changes of too small jump length amplitude. In the intermediate regime a scenario emerges resembling qualitatively the *PSD* patterns as observed empirically in the EEG signals, displaying rates of synchronization changes on the time scale of seconds. For that scenario, $k$-medoids clustering of the transient phase synchronization networks using the *PSD* matrix as a (correlation) distance metric reveals three differentiated clusters of phase synchronization networks, i.e., three distinct network "states" which persist and reappear during the 10 sec interval of the simulation. Mapping *PSD* to a 3-dimensional trajectory in a space spanned by correlations of each synchronization network at time window $t$ to one of the three clusters' medoids, illustrates PSD dynamics as a stochastic exploration of the space of synchronization network "states".

## Discussion

Changes in brain-signal variability across the lifespan—spanning childhood, adulthood, and older adulthood—illuminate normative development and reveal windows of resilience or vulnerability that may serve as biomarkers of cognitive health or frailty. Yet even the most richly analyzed datasets cannot, on their own, expose the principles that connect neuronal activity to the computations it supports [79]. Deeper insight arises when neuroimaging and advanced analytics are integrated with computational modeling anchored in a mechanistic theory of brain function [12].

Adopting such a lifespan framework, we analyzed resting-state and task-evoked EEG from four age groups (9–75 years) using univariate statistics and time-resolved phase-synchrony metrics. Across both states we found two co-existing age trajectories: a monotonic decline and a pronounced inverted U-shaped trend. Follow-up simulations showed that a network of coupled phase oscillators can reproduce these non-stationary synchrony patterns, offering a mechanistic lens on how neural variability is tuned by development, maturation, and aging. We now detail the implications of these findings.

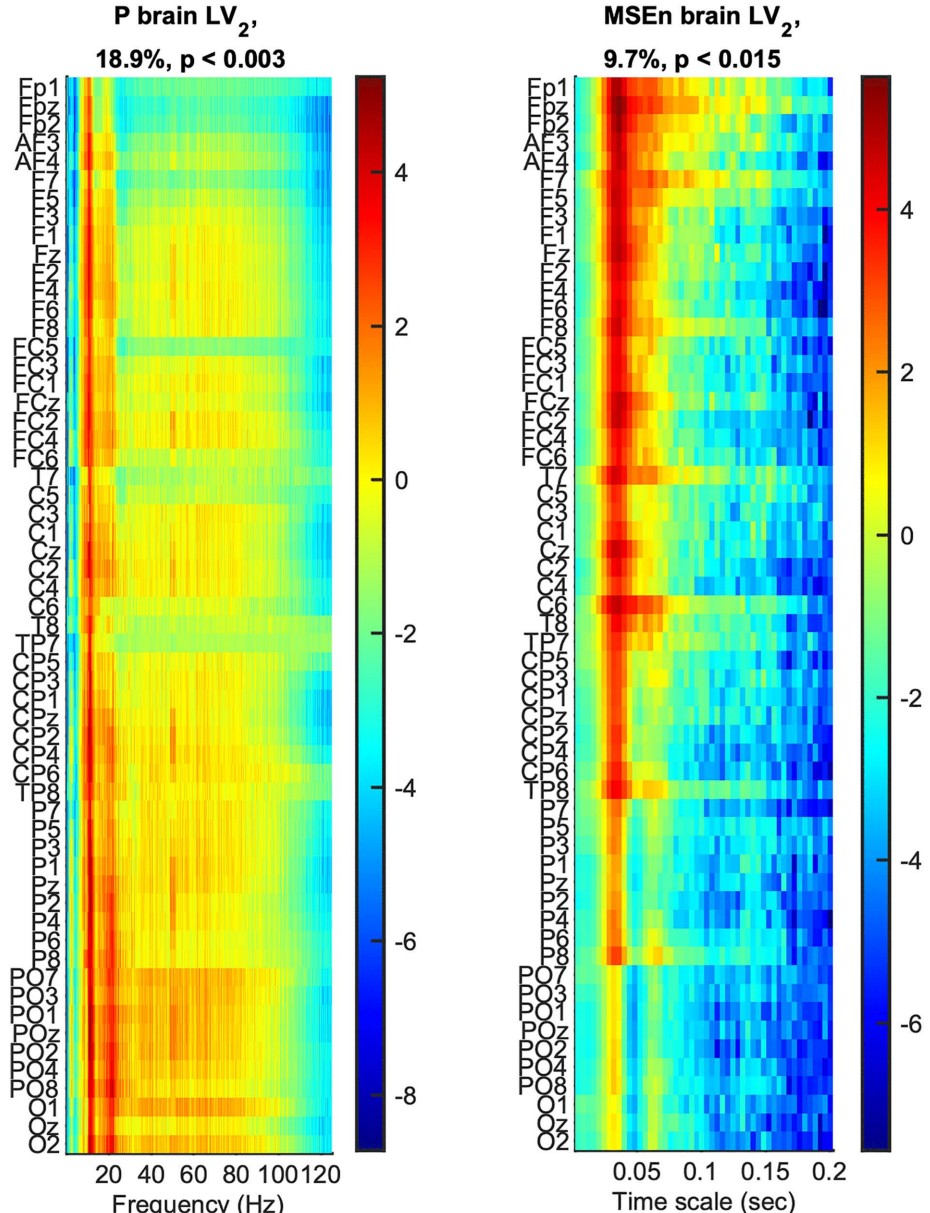

**Fig 7. Second brain latent variable of univariate variability metrics: U-shape component.** The panels show how much each data *element*, i.e., a metric's data point, covaries with the task contrasts of Fig 6 that correspond to the (inverted) U-shape-like change across lifespan for *P* and *MSEn*, in terms of bootstrap ratios. Axes and other conventions are identical to Fig 5. Positive (negative) values signify points with positive (negative) correlation with the task latent variables of Fig 6, i.e., with values that were higher (lower) for '*OC*' and '*YA*' ('*YC*' and '*OA*', respectively).

## Brain dynamics as revealed by multiscale EEG fluctuations

Independent of the experimental condition (rest or task), the univariate metrics–Power spectrum (P), standard deviation (σ), Multiscale Entropy (MSE), normalized Multiscale Entropy (MSEn)–revealed a clear distinction between children and adults, with the suggestion of a monotonic lifespan change from younger to older ages. This included a decrease in power and in the amplitude of fluctuations for slow frequencies (below 10 Hz) and longer timescales, respectively. These

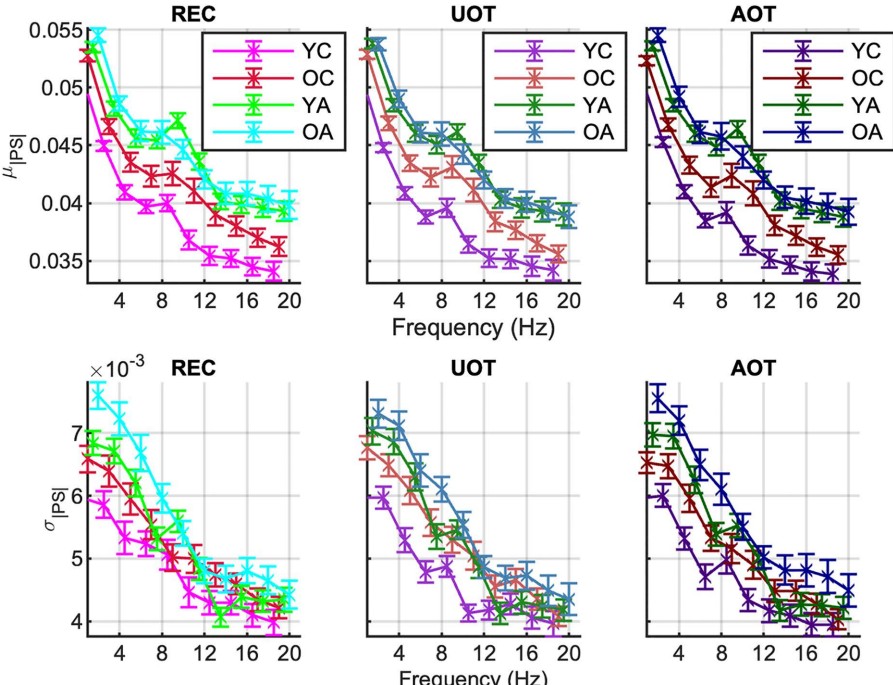

**Fig 8. Group and condition's means and standard error intervals of the metrics of the phase synchronization norm.** The sample mean $\mu_{|PS|(f)}$ (top row) and standard deviation $\sigma_{|PS|(f)}$ (bottom row) of the *phase synchronization norm* ($|PS(t,f)|$), computed across time for each frequency (x-axes), are shown. Group and condition colors correspond to those in Fig 3. Columns correspond to conditions of increasing attentional demands from left to right ('*REC*', '*UOT*', '*AOT*').

task-independent effects were also reflected in reduced complexity of EEG fluctuations with age resulting from significantly lower entropy at longer timescales that mirror slower processes (> 20 ms for MSE; > 50 ms for MSEn) in the adults compared to the children's groups. These changes were most pronounced at frontal electrode sites. Interestingly, for two of the studied metrics, *P* and *MSEn*, the dominant monotonic trend co-existed with a nonlinear trend (inverted U-shaped) opposing older children and younger adults to younger children and older adults. The inverted U trend concerned an intermediate range of frequencies (6–24 Hz) and timescales (20–50 ms), with more power and higher entropy for the intermediate age groups compared to the youngest children and the oldest adults. This result was most prominent at the posterior channels for P and the fronto-central channels for MSEn.

The observed decrease in slow-frequency power (below 10 Hz) and the reduced complexity at longer timescales align with previously reported developmental EEG patterns, with delta and theta power declining from childhood to adulthood, along with brain maturation processes [33,80]. The frontal predominance of these changes is particularly consistent with the late development of the prefrontal cortex, which continues maturing well into the third decade of life. The finding of decreased multiscale entropy at longer timescales (>20–50ms) in adults versus children completes previous findings that focused on shorter timescales [28,59] and concluded an increase in brain signal variability with development. Here, we show that EEG variability in terms of magnitude ($\sigma$) and complex time-structure is lower in children than in adults. This could be linked to a possible shift in how the brain processes information (less locally and more distributed) with age [45].

A noteworthy feature of our data is that monotonic decline and inverted U-shaped trajectories coexist within the same EEG recordings. Such duality has been predicted by theory and documented with MRI connectivity but rarely

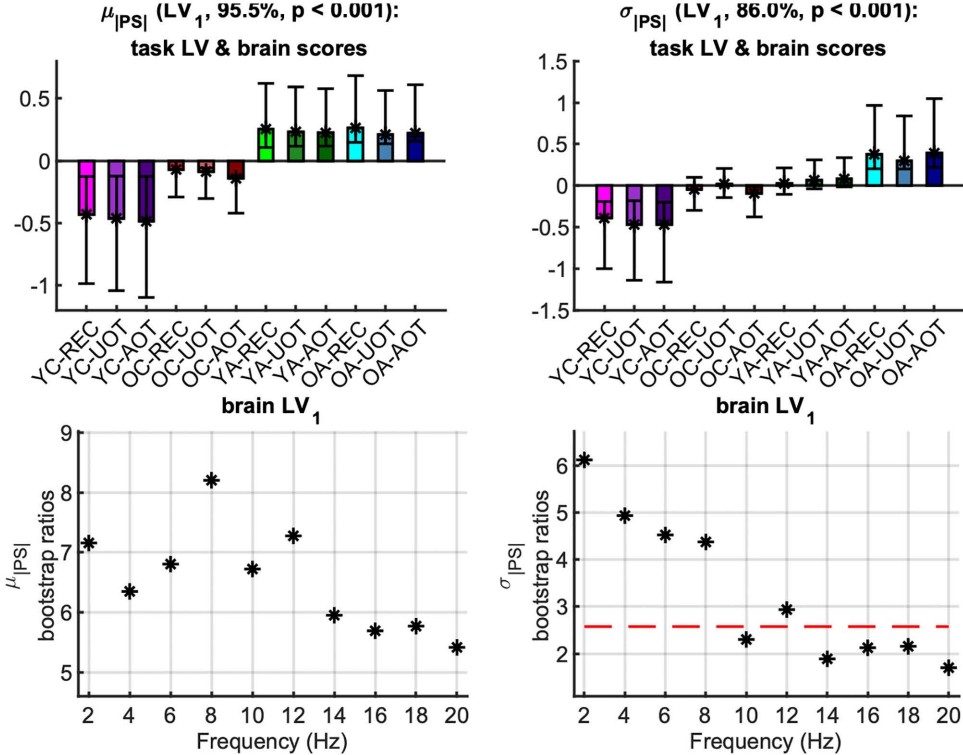

**Fig 9. First task latent variable, normalized brain scores and brain latent variable of the phase synchronization norm: monotonic component.**
*Top row:* Each panel shows the weights of the first *task latent variable* (asterisks), and the normalized brain scores with 95% confidence intervals (bars and error bars) *variable* of $\mu_{|PS|(f)}$ (left) and $\sigma_{|PS|(f)}$ (right). Color and other conventions are identical to [Fig 4]. We observe generally monotonically increasing effects with age that reliably contrast adults to children for $\mu_{|PS|(f)}$, and old adults to young children, with old children and young adults being somewhere in between, for $\sigma_{|PS|(f)}$. *Bottom row:* Each panel shows the bootstrap ratios of the brain latent variable of $\mu_{|PS|(f)}$ (left) and $\sigma_{|PS|(f)}$ (right; note the threshold of the two-tailed 99th percentile with the red dotted line). All bootstrap ratios are positive, i.e., they correspond to elements that correlate positively with contrasts of the top row, i.e., where adults have higher values, for $\mu_{|PS|(f)}$, and where old adults have higher values for $\sigma_{|PS|(f)}$.

verified in lifespan EEG. Here, the inverted U-shaped pattern emerges at intermediate frequencies (≈ 6–24 Hz) and timescales of 20–50 ms, hinting that specific network operations reach optimal efficiency in young adulthood before tapering with age. One plausible contributor is the transient surge in posterior beta activity observed during cognitive maturation, whose scalp distribution overlaps the classic posterior-dominant alpha topography [81]. Finally, we should note that the common trends among all four univariate measures for LV1, as well as for the LV2 for *P* and *MSEn*, could reflect linear autocorrelation properties of the EEG signals, rendering them partially redundant. This is true mainly for *P* and *σ*, since *MSE* and *MSEn* have been shown to be more sensitive to nonlinear properties of signals [41,43,71].

## Brain dynamics as revealed by EEG phase synchronization dynamics

The analysis of the EEG synchronization dynamics afforded a richer account of the lifespan changes as reflected by the differences between the studied age groups across all three task-conditions (REC, UOT, AOT). We found first that, across time and for each frequency, the mean and the variability of the magnitude of the phase synchronization networks increased with age (cf. [35]). This was the case for all frequencies, but more reliably so for the lower ones (<10 Hz). The mean of the phase synchronization norm contrasted between children and adults, while its standard deviation opposed

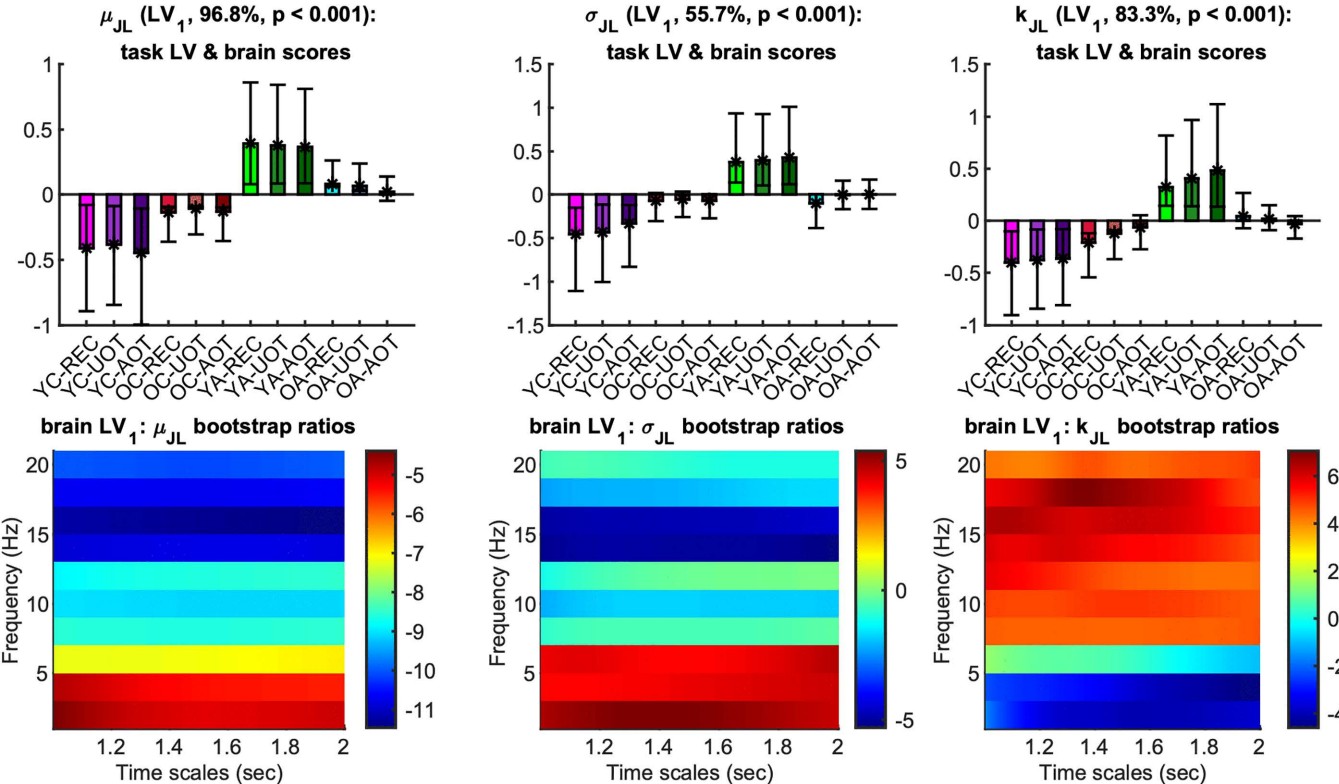

**Fig 10. First task latent variable, normalized brain scores and brain latent variable of the phase synchronization dynamics metrics: U-shape component.** *Top row:* Each panel shows the weights of the first *task latent* (asterisks), and the normalized brain scores with 95% confidence intervals (bars and error bars) *variable* of $\mu_{JL(\tau,f)}$ (left), $\sigma_{JL(\tau,f)}$ (center) and $k_{JL(\tau,f)}$ (right). Color and other conventions are identical to Fig 4. We observe for all metrics inverted U-shape components with age, with reverse effects for development and aging that reliably contrast young adults to children, with old adults being somewhere in between. *Bottom row:* Each panel shows the bootstrap ratios of the brain latent variable of $\mu_{JL(\tau,f)}$ (left), $\sigma_{JL(\tau,f)}$ (center) and $k_{JL(\tau,f)}$ (right). Positive (negative) values correspond to elements that correlate positively (negatively) with contrasts of the top row, i.e., where young adults (children) have higher (lower) values (respectively).

young children to older adults, with older children and young adults being undistinguishable from the two former groups. As with the above-discussed univariate metrics, these two descriptive bivariate metrics of phase synchronization point out a monotonic change in brain dynamics, with a stronger and more variable functional (phase synchronization) connectivity when going from early childhood to aging.

Moving to the analysis of phase synchronization dynamics (PSD), we could extract a different trajectory of how age affects the functioning of the brain by altering the topology of the phase synchronization networks following a nonlinear inverted U-like trend opposing development to aging. Independently of the task condition, younger adults were reliably distinguished from all the three other age groups for all the jump length distribution parameters. Namely, they showed a lower mean jump length for all frequencies, suggesting a slower switching dynamic compared to the children and older adults. They were also found to be more variable and less leptokurtic in their switching dynamics than the other groups for low frequency bins (2–6 Hz), and the reverse for the higher frequencies (8–20 Hz). Accordingly, we can conclude that development and aging modulate the PSD in opposite directions, thereby modifying the expression of the transient fluctuations of the functional (phase synchronization) networks. This discriminating transient dynamic can be captured best through measures that do not rely on or assume stationarity of the brain's functional connectivity. Our results are highly consistent with [61], reinforcing the inverted U-shaped model of EEG phase synchronization across the lifespan and the value

of dynamic, non-stationary analysis. Our additional focus on the detailed transient switching dynamics (PSD) provides a complementary perspective, offering finer granularity on how development and aging shape the flexibility and variability of brain network connectivity. The finding that younger adults show distinct switching dynamics–slower, more variable switching at low frequencies and less variable at higher frequencies–suggests an optimal dynamic regime in young adulthood. This aligns with theoretical frameworks proposing that the mature healthy brain operates in a metastable state balancing integration and segregation [35,38,81]. However, we should mention that future work should address a limitation in our present analysis, namely the restriction to frequencies up to 20 Hz.

## Modeling the synchronization dynamics in a network of coupled phase oscillators

Earlier work has shown that large-scale "cascades" of spontaneous activity—bursts that sweep through distributed networks—are detectable in both EEG and fMRI and reorganize systematically with age [9,14,82–85]. These phenomena have been reproduced in whole-brain simulations, confirmed in animal models, and observed with simultaneous multimodal recordings. Cascades tend to cluster during epochs when the underlying dynamics are quasi-stationary, implying that they both signal and stabilize favorable network states.

We capture an analogous signature here, albeit in a deliberately pared-down form: our analyses focus exclusively on phase relationships. Excluding amplitude inevitably misses waxing-and-waning envelope modulations, but a band-limited, phase-only approach is standard in EEG research and maps cleanly onto networks of Kuramoto oscillators. Within this framework, cascades appear as brief windows of heightened inter-regional coherence—transient "tiles" in the temporal mosaic of brain activity. fMRI studies link richly cascaded, or "fluid," dynamics to robust cognitive performance; this fluidity wanes with advancing age and correlates with declining cognition [14]. Our EEG findings echo that pattern, showing that time-resolved phase synchrony provides an accessible proxy for the same health-related dynamical regime.

From a dynamical-systems perspective, our oscillator model provides a controlled reference for interpreting the synchronization metrics used in the empirical analysis. It is expected that very weak coupling yields largely independent oscillators with little synchronization, whereas very strong coupling leads to rigid phase locking. However, central to the present study, the model illustrates how these different coordination regimes express themselves in the statistical signatures of network switching quantified in the data. The simulations demonstrate that only an intermediate coupling regime produces a metastable landscape in which multiple partially synchronized network configurations coexist and are intermittently explored. This regime generates slow, structured transitions among network states and a broad distribution of switching events, precisely the type of behavior captured by the jump-length statistics and topology-based synchronization dynamics measures applied to the EEG recordings.

The oscillator model used here is intentionally phenomenological and not designed to map each parameter onto a single specific biological process. Instead, the parameters represent effective dynamical control variables governing large-scale synchronization. The connectivity matrix constitutes the only explicitly anatomical component of the model, as it is derived from diffusion-MRI tractography and therefore reflects the structural white-matter scaffold constraining inter-regional communication. By contrast, the global coupling parameter controls the overall strength of interactions across this scaffold and should be interpreted as an aggregate measure of effective inter-regional communication efficacy. Such efficacy may arise from multiple biological sources, including synaptic gain, neuromodulatory tone, excitation–inhibition balance, or conduction reliability, rather than a single identifiable mechanism and illustrates the degeneracy present in multiscale systems. Similarly, the local stability parameter regulates how strongly each oscillator tends to remain in, or depart from, a phase-locked configuration. At a biological level, this corresponds broadly to regional excitability and the tendency of neuronal populations to engage in synchronized activity, again reflecting a compound effect of multiple microscopic processes. A characterization of neurodegeneracy and the mapping of specific biological parameters onto reduced dynamical parameters are objectives of virtual brain twin approaches [86–88] and pave the road to applications in personalized medicine.

## Conclusion

Our results show that both the amplitude of EEG fluctuations and the tempo of phase synchrony change with age along two distinct trajectories that hold independent of task demands. Aggregate, "static" indices exhibit primarily a roughly linear decline from childhood to older adulthood, explaining most of the variance, whereas transient, time-resolved metrics of network switching follow a pronounced inverted U-shaped trajectory, peaking in young adulthood. Detailed jump-length analyses (mean, variance, kurtosis across frequencies) reveal that young adults execute slower, more selective transitions between functional states, consistent with an optimally tuned balance between stability and flexibility. Children appear to be learning this control regime, while older adults show a drift toward faster, less discriminating switches, suggesting a loss of the optimized switching dynamics established at the earlier stage of life.

An inverted U-shaped trend component was also observed for power/MSEn changes. This likely reflects a similar underlying lifespan reconfiguration of large-scale networks—maximal distributed, flexible coordination in young adult-hood—seen from a "local signal" perspective (power/MSEn) and a "global network dynamics" perspective (phase-synchrony state switching). On the one hand, the changes in P and MSEn in the 6–24 Hz range can be interpreted as reflecting a lifespan peak in the local repertoire and temporal differentiation of oscillatory activity, particularly in posterior and fronto-central regions [29]. On the other hand, and in comparable frequency bands, PSD changes suggest a lifespan peak in the exploration of functional network states, with young adults showing a controlled yet flexible exploration regime. The convergence of these patterns suggests that maximal local complexity (indexed by power/MSEn) around mid-frequencies could be a prerequisite for maximal flexibility and efficiency of larger-scale synchronization dynamics, both of which peak in young adulthood and decline at the two ends of the lifespan [16]. In other words, the optimal level of local variability in young adults constitutes an oscillatory drive that enables the brain to sample many phase-synchronization configurations without either undersampling (rigidity) or oversampling (instability).

Overall, these electrophysiological patterns mirror the dual monotonic and inverted U-shaped trends long reported in structural and functional MRI, implying a common mechanistic underpinning. Within the framework of Structured Flows on Manifolds, our findings link micro-scale neural dynamics to macro-scale network architecture, providing a coherent view across imaging modalities and timescales. We note that sample size can be considered a limitation of the present cross-sectional study, particularly for population-level inferences about lifespan development and aging, even though we sought to mitigate its impact on the statistical results by relying on permutation and bootstrap procedures; this caveat should be borne in mind when extrapolating our findings beyond the sampled cohorts. A critical next step is to extend this approach to larger samples while including simultaneous behavioral recordings, allowing us to trace how brain and behavior co-evolve across the human lifespan.

## Supporting information

**S1 Fig. Connectivity used for simulations.**
(DOCX)

**S2 Fig. Group and condition means of $\mu_{JL(\tau,f)}$.** Groups are arranged in rows (age increases from top to bottom) and conditions in columns (attentional demands increase from left to right, i.e., '*REC*', '*UOT*', '*AOT*'). Time scale in seconds for all x-axes and frequency in Hz for y-axes. Note the transient effect for approximately the first second that corresponds to the length of the –fully overlapping- sliding window of phase synchronization computation. Only data for timescales in the range 1.004-2 sec, almost 1 sec after the length of the overlapping window, were used for further statistical analysis.
(DOCX)

**S3 Fig. Group and condition t-score of $\mu_{JL(\tau,f)}$.** Figure arrangement and conventions similar to S2 Fig.
(DOCX)

**S4 Fig. Group and condition means of $\sigma_{JL(\tau,f)}$.** Figure arrangement and conventions similar to S2 Fig.
(DOCX)

**S5 Fig. Group and condition t-score of $\sigma_{JL(\tau,f)}$.** Figure arrangement and conventions similar to S2 Fig.
(DOCX)

**S6 Fig. Group and condition means of $k_{JL(\tau,f)}$.** Figure arrangement and conventions similar to S2 Fig.
(DOCX)

**S7 Fig. Group and condition t-score of $k_{JL(\tau,f)}$.** Figure arrangement and conventions similar to S2 Fig.
(DOCX)

**S8 Fig. Second task latent variable, normalized brain scores and brain latent variable of the phase synchronization dynamics metrics.** *Top row:* Each panel shows the weights of the second *task latent* (asterisks), and the normalized brain scores with 95% confidence intervals (bars and errorbars) *variable* $\sigma_{JL(\tau,f)}$ (left) and $k_{JL(\tau,f)}$ (right). Color and other conventions are identical to Fig 10 of the main text. This LV is significant ($p < 0.003$ for $\sigma_{JL(\tau,f)}$ and $p < 0.006$ for $k_{JL(\tau,f)}$) and explained approximately 33.3% and 6.8% of the covariance, respectively It contrasts '*OC*' and '*OA*' to '*YC*' and '*YA*' –in general- with statistical reliability (non-overlapping confidence intervals) for $\sigma_{JL(\tau,f)}$. Instead, the contrast described by the respective LV for $k_{JL(\tau,f)}$ is not reliable at all. *Bottom row:* Each panel shows the bootstrap ratios of the brain latent variable of $\sigma_{JL(\tau,f)}$ (left) and $k_{JL(\tau,f)}$ (right). Positive (negative) values correspond to elements that correlate positively (negatively) with contrasts of the top row, i.e., where that young adults (children) have higher (lower) values, respectively. Only values at 8 Hz for $\sigma_{JL(\tau,f)}$, and at 16 and 20 Hz for a short range of scales around 1.6 sec and 1.85 sec respectively, for $k_{JL(\tau,f)}$, have bootstrap ratios with absolute values greater than 2.5758. Therefore, these contrasts are statistically unreliable in general, and are not discussed in the main text.
(DOCX)

**S1 Table. The 19 EEG electrodes used for the phase synchronization computation in the original 10–10 nomenclature and their mapping to 10–20 system equivalents.**
(DOCX)

## Author contributions

**Conceptualization:** Dionysios Perdikis, Rita Sleimen-Malkoun, Viktor Jirsa.

**Data curation:** Viktor Müller.

**Formal analysis:** Dionysios Perdikis, Rita Sleimen-Malkoun.

**Funding acquisition:** Rita Sleimen-Malkoun, Viktor Müller, Viktor Jirsa.

**Investigation:** Dionysios Perdikis, Rita Sleimen-Malkoun, Viktor Müller.

**Methodology:** Dionysios Perdikis, Rita Sleimen-Malkoun, Viktor Müller, Viktor Jirsa.

**Resources:** Viktor Müller.

**Software:** Dionysios Perdikis.

**Visualization:** Dionysios Perdikis.

**Writing – original draft:** Dionysios Perdikis, Rita Sleimen-Malkoun.

**Writing – review & editing:** Dionysios Perdikis, Rita Sleimen-Malkoun, Viktor Müller, Viktor Jirsa.

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
