## [Decision Letter · Decision Letter 0]

8 Dec 2025

PCOMPBIOL-D-25-01331

Developmental and Aging Changes in Brain Network Switching Dynamics Revealed by EEG Phase Synchronization

PLOS Computational Biology

Dear Dr. Perdikis,

Thank you for submitting your manuscript to PLOS Computational Biology. After careful consideration, we feel that it has merit but does not fully meet PLOS Computational Biology's publication criteria as it currently stands. Therefore, we invite you to submit a revised version of the manuscript that addresses the points raised during the review process.

We sincerely regret that the review process has been so long. Indeed, it was particularly challenging to find qualified reviewers for your manuscript. As you can see from the reviews, they were quite enthusiastic with your results, though with important points for revision.

We look forward to receiving your revised manuscript.

Kind regards,

Lyle J. Graham

Section Editor

PLOS Computational Biology

**Journal Requirements:**

1) We have noticed that you have uploaded Supporting Information files, but you have not included a list of legends. Please add a full list of legends for your Supporting Information files after the references list.

2) Kindly revise your competing statement in the online submission form to align with the journal's style guidelines: 'The authors declare that there are no competing interests.'

**Reviewers' comments:**

Reviewer's Responses to Questions

**Comments to the Authors:**

Reviewer #1: I think this is a high-quality and impactful work. The results offer new insights into brain aging, and help bridge the gap between neuroimaging and the mechanistic research at the micro- and mesoscales. The authors achieve this through robust time-series analyses of EEG data from a sample with an exceptionally wide lifespan coverage (9–75 years). Using a set of metrics based on activity and phase synchronization, they identify distinct age-related patterns of monotonic trends along with features that show inverted U-shaped trends, helping to dissociate brain development and aging effects. The paper overall is well written and the concepts are clearly articulated. I particularly appreciated the effort to include simplified explanations alongside complex concepts, these helped a lot with readability. My comments are only minor and primarily focus on further improving the clarity of the work. I believe this paper would be an impactful contribution and a good fit for PLOS Computational Biology.

1. I understand there were three different scans (rec, uot, aot) from each participant, and that they were considered simultaneously in the PLS analyses. Could you provide some assurance that the task-related effects didn’t confound the age-related ones? Would the identified age-related trends persist if multiple instances of PLS were run separately for the different tasks? (no need to carry out these analyses if the answer is simple)

2. Relatedly, did you notice any trends with respect to the task? Do the uot/aot tasks represent a heavier cognitive load? If so, does this show up in any of the metrics you looked at?

3. I think the presented simulations were valuable in proposing mechanistic hypotheses, and I felt these results could be discussed in a bit more detail to better motivate follow-up experimental work. Particularly, a short discussion of how the model parameters correspond to specific mechanisms would be useful.

4. Do you have any thoughts on how the inverted U-shaped trends in power/MSEn may relate to the inverted U-shaped trend in phase synchronization dynamics? If I missed this in the text, I’d appreciate a pointer to it.

5. I think the line between what literature already exists and what this work is adding could be more explicitly defined in the introduction.

6. For the phase synchronization analyses, on what criteria were the 19 regions selected, and where can I find a list of these regions?

7. The PLS step and its justification make sense, but it took me several reads to understand. I’d suggest considering a figure with a schematic explaining this step. I leave this to your judgment. It could potentially be added as a panel in Figure 1, if there’s space.

8. Figure 1 appears low resolution on my end, even when accessing the raw TIF file. If this is inherent to the figure, I’d suggest regenerating it in higher resolution, some details are hard to read.

9. Despite the caption referencing it, I couldn’t locate the “thick dotted black line” in Figure 1, panel F.

10. On Figures 2 and 7, it would be helpful to include x-tick labels in every row. Having to constantly look across the figure makes it harder to navigate.

11. The percentages shown in Figures 2–6 and 8–9 seem overly precise in my opinion and are a bit distracting

12. I couldn’t find a definition for the acronym “LV”; I assumed it stands for latent variable, but it would be good to add this explicitly.

Reviewer #2: This manuscript examines age trajectories in several EEG features, as indicators of changes in state dynamics. A key finding is that both monotonic and inverted-U (non-monotonic) trajectories were found, coexisting within the feature variance, and consistent with multiple system dynamics changing independently across age. The manuscript is overall clearly presented, methods are rigorous, and the perspective is innovative, offering a multivariate perspective on brain state dynamics across age, and including a simulation model based on oscillator coupling, that could have broad interest. Some concerns in the current versions are outlined below, and include consideration of confounds, as well as sparsity in some analytical details and rationale that could affect the impact or interpretation of results.

Considerations:

1. The sample size of n=111 is not large for population scale inferences about development, especially because it is cross-sectional not longitudinal. This is a known limitation of population studies, and will of course affect the statistics (e.g., the generalizability of the cross-covariance PLS matrices, based on n=24-31 individuals per group). This ought to be discussed in the caveats.

2. The pre-processing for EEG is omitted, with references provided instead. Given that cleaning protocols may impact the features that are at the heart of the analyses presented, these protocols ought to be presented in full. This is particularly important because whether and how much there are confounding signals contributing to the results, can not be evaluated without such information (or perhaps relate to why the EO condition is different and omitted).

3. It is somewhat curious that EO was omitted due to a “different power profile”. This is notable because this condition is more akin to task than the EC condition. Given that the groups effects are common across tasks, this suggests the possibility that some odd confounding property of the signal could be contributing to the similarity between EC and task. An example is amount of blinks in the data or noise. Perhaps the task differences could be more fully justified and explained in the supplemental materials.

4. Relatedly, it would be helpful to see assessments of the data quality, or the results of data cleaning. It is always a concerns that the age effects are because children move more or blink less, affecting the entire time series via confound signals not neural state changes.

5. Finally, also related to 2-4 above, it would be valuable to know how these different features relate to one another? Would variance and MSE be considered redundant through mathematical relationships? How about phase coupling, or relationships between PSD and MSE? Note that across features the first LV shows a monotonic change and the second LV shows a non-monotonic change - would this be a side effect of these measures all being related to one another?

6. The explanations of core features (e.g., MSE, extended MSE) and concepts (stationarity & why PSD is non-stationary) in the introduction could be expanded to make the rationale and predictions more explicit. The significance of the simulation and choice of coupled oscillations is similarly not entirely clear. Why is this particular model valuable?

7. Lines 666-668 state that there exists a differentiation between monotonic and non-monotonic patterns as synonymous with static and non-static measures but this is not quite in line with the findings, which do also find (in LV2) non-monotonic effects for “static” measures (MSE, P).

Minor notes:

- The figures lack resolution - e.g., figure1 text is hard to read in places.

- The functions used for extraction of features are inconsistently documented (listed in Fig 1 legend for some, but not all, and not in text).

Reviewer #3: The authors present a fascinating study on lifespan trajectories of neural variability and synchronization using EEG and computational modeling. The integrated empirical-computational approach is a significant strength, and the findings regarding distinct monotonic and inverted U-shape trajectories are novel and of potential broad interest. The claim that metastable coupling underlies the optimal dynamics observed in young adults is compelling. However, several major issues need to be addressed before the manuscript can be considered for publication.

Major Concerns:

1.The restriction of the EEG analysis to the 2-20 Hz range is a significant limitation that requires thorough justification. The authors must provide a clear, a priori rationale for excluding other relevant frequency bands. Why were the other high-frequency oscillations excluded?How do the authors ensure that the key lifespan trajectories they describe are not confounded by or do not manifest in these excluded frequencies? For instance, age-related changes in the power of other frequency bands are well-documented in the literature.

2.The manuscript states that analyses were performed on 19 electrodes, but it lacks a critical methodological detail: how and why these specific 19 electrodes were chosen from the original high-density (58-electrode) setup. Please specify the exact selection criteria (e.g., based on a standard 10-10 system subset, covering specific brain regions, or selected based on data quality).

3.The computational model is a key component of the manuscript, but its current validation appears incomplete.The authors claim the model reproduced "the empirical combination of reduced low-frequency variability." However, the results seem to focus solely on replicating the phase-synchrony dynamics. The model's ability to replicate the univariate empirical findings (e.g., the monotonic decline in slow-frequency power, variance, and complexity) must be explicitly demonstrated. Does the model's output show the same age-dependent decline in these low-frequency measures?

4.The model is used to explain differences between young adults, children, and older adults. However, the results lack a direct, visual comparison of the model's output for these specific groups against the empirical data. The authors should present a figure equivalent to Figure 9 (which nicely contrasts the empirical data across age groups) for the model's output. This would allow for a direct visual assessment of how well the model captures the distinct dynamical profiles of children and older adults compared to young adults.

Minor Concerns:

1.In Figure 1, the x-axes in several panels lack units (e.g., likely "Hz" for frequency or "ms" for time). Please add the appropriate units to all axes for clarity.

2.In Figure 10, Panels A and B are mentioned in the figure legend but are not referenced in the main text. The text should guide the reader through all panels of the figure.

**Have the authors made all data and (if applicable) computational code underlying the findings in their manuscript fully available?**

The PLOS Data policy requires authors to make all data and code underlying the findings described in their manuscript fully available without restriction, with rare exception (please refer to the Data Availability Statement in the manuscript PDF file). The data and code should be provided as part of the manuscript or its supporting information, or deposited to a public repository. For example, in addition to summary statistics, the data points behind means, medians and variance measures should be available. If there are restrictions on publicly sharing data or code —e.g. participant privacy or use of data from a third party—those must be specified.requires authors to make all data and code underlying the findings described in their manuscript fully available without restriction, with rare exception (please refer to the Data Availability Statement in the manuscript PDF file). The data and code should be provided as part of the manuscript or its supporting information, or deposited to a public repository. For example, in addition to summary statistics, the data points behind means, medians and variance measures should be available. If there are restrictions on publicly sharing data or code —e.g. participant privacy or use of data from a third party—those must be specified.requires authors to make all data and code underlying the findings described in their manuscript fully available without restriction, with rare exception (please refer to the Data Availability Statement in the manuscript PDF file). The data and code should be provided as part of the manuscript or its supporting information, or deposited to a public repository. For example, in addition to summary statistics, the data points behind means, medians and variance measures should be available. If there are restrictions on publicly sharing data or code —e.g. participant privacy or use of data from a third party—those must be specified.requires authors to make all data and code underlying the findings described in their manuscript fully available without restriction, with rare exception (please refer to the Data Availability Statement in the manuscript PDF file). The data and code should be provided as part of the manuscript or its supporting information, or deposited to a public repository. For example, in addition to summary statistics, the data points behind means, medians and variance measures should be available. If there are restrictions on publicly sharing data or code —e.g. participant privacy or use of data from a third party—those must be specified.

Reviewer #1: Yes

Reviewer #2: Yes

Reviewer #3: Yes

PLOS authors have the option to publish the peer review history of their article (what does this mean?). If published, this will include your full peer review and any attached files.). If published, this will include your full peer review and any attached files.). If published, this will include your full peer review and any attached files.). If published, this will include your full peer review and any attached files.

...

Reviewer #1: No

Reviewer #2: No

Reviewer #3: No

**Figure resubmission:**

**Reproducibility:**



---

## [Decision Letter · Decision Letter 1]

29 Mar 2026

Dear Dr. Sleimen-Malkoun,

We are pleased to inform you that your manuscript 'Developmental and Aging Changes in Brain Network Switching Dynamics Revealed by EEG Phase Synchronization' has been provisionally accepted for publication in PLOS Computational Biology.

You can also use this step to take into consideration the comments of Reviewer #1 below.

Best regards,

Lyle Graham

Section Editor

PLOS Computational Biology

Reviewer's Responses to Questions

**Comments to the Authors:**

Reviewer #1: I appreciate the authors’ responses and believe the revisions have further strengthened this already strong and impactful work. In particular, the added discussion points are helpful and will make it easier for readers to interpret the results and consider how to build on this work.

I have one comment related to the PLS-based analysis. My earlier comments (1. and 2.) did not imply that PLS itself tested for any contrasts, but rather referred to the subsequent tests on the latent variables. While the response clarifies that the dominant LVs don’t show strong task-related effects (rec, uot, aot), and this was also originally suggested by Figures 3, 5, 8, and 9, I still couldn’t find an explicit mention of this in the results or discussion.

The absence of any explicit discussion of task-related effects seems odd given that the three task conditions were explicitly distinguished and included in the PLS analysis and then subsequently compared through the latent variables. Since these analyses were performed, I think their outcome should be discussed with clarify and explicitness. Even if such effects "were not central to the main research question" (which seems inconsistent with the stated focus on neurocognitive status), both the presence of trends and the lack thereof with respect to cognitive load are important to the field and would be of interest to readers.

It’s possible I may have overlooked this in the manuscript, and in any case, I don’t view this as affecting the overall impact of the work. I offer this comment in a constructive spirit and leave it to the authors’ discretion. I appreciate the opportunity to review this well-written and interesting work.

**Have the authors made all data and (if applicable) computational code underlying the findings in their manuscript fully available?**

The PLOS Data policy requires authors to make all data and code underlying the findings described in their manuscript fully available without restriction, with rare exception (please refer to the Data Availability Statement in the manuscript PDF file). The data and code should be provided as part of the manuscript or its supporting information, or deposited to a public repository. For example, in addition to summary statistics, the data points behind means, medians and variance measures should be available. If there are restrictions on publicly sharing data or code —e.g. participant privacy or use of data from a third party—those must be specified.requires authors to make all data and code underlying the findings described in their manuscript fully available without restriction, with rare exception (please refer to the Data Availability Statement in the manuscript PDF file). The data and code should be provided as part of the manuscript or its supporting information, or deposited to a public repository. For example, in addition to summary statistics, the data points behind means, medians and variance measures should be available. If there are restrictions on publicly sharing data or code —e.g. participant privacy or use of data from a third party—those must be specified.requires authors to make all data and code underlying the findings described in their manuscript fully available without restriction, with rare exception (please refer to the Data Availability Statement in the manuscript PDF file). The data and code should be provided as part of the manuscript or its supporting information, or deposited to a public repository. For example, in addition to summary statistics, the data points behind means, medians and variance measures should be available. If there are restrictions on publicly sharing data or code —e.g. participant privacy or use of data from a third party—those must be specified.requires authors to make all data and code underlying the findings described in their manuscript fully available without restriction, with rare exception (please refer to the Data Availability Statement in the manuscript PDF file). The data and code should be provided as part of the manuscript or its supporting information, or deposited to a public repository. For example, in addition to summary statistics, the data points behind means, medians and variance measures should be available. If there are restrictions on publicly sharing data or code —e.g. participant privacy or use of data from a third party—those must be specified.

Reviewer #1: Yes

PLOS authors have the option to publish the peer review history of their article (what does this mean?). If published, this will include your full peer review and any attached files.). If published, this will include your full peer review and any attached files.). If published, this will include your full peer review and any attached files.). If published, this will include your full peer review and any attached files.

...

Reviewer #1: No

---

## [Editor Report · Acceptance letter]

PCOMPBIOL-D-25-01331R1

Developmental and Aging Changes in Brain Network Switching Dynamics Revealed by EEG Phase Synchronization

Dear Dr Sleimen-Malkoun,

I am pleased to inform you that your manuscript has been formally accepted for publication in PLOS Computational Biology. Your manuscript is now with our production department and you will be notified of the publication date in due course.

With kind regards,

Judit Kozma
